# Structural basis of ribosomal 30S subunit degradation by RNase R

Lyudmila Dimitrova-Paternoga[1], Sergo Kasvandik[2], Bertrand Beckert[3], Sander Granneman[4], Tanel Tenson[2], Daniel N. Wilson[1 ✉] & Helge Paternoga[1 ✉]

Protein synthesis is a major energy-consuming process of the cell that requires the controlled production[1–3] and turnover[4,5] of ribosomes. Although the past few years have seen major advances in our understanding of ribosome biogenesis, structural insight into the degradation of ribosomes has been lacking. Here we present native structures of two distinct small ribosomal 30S subunit degradation intermediates associated with the 3′ to 5′ exonuclease ribonuclease R (RNase R). The structures reveal that RNase R binds at first to the 30S platform to facilitate the degradation of the functionally important anti-Shine–Dalgarno sequence and the decoding-site helix 44. RNase R then encounters a roadblock when it reaches the neck region of the 30S subunit, and this is overcome by a major structural rearrangement of the 30S head, involving the loss of ribosomal proteins. RNase R parallels this movement and relocates to the decoding site by using its N-terminal helix-turn-helix domain as an anchor. In vitro degradation assays suggest that head rearrangement poses a major kinetic barrier for RNase R, but also indicate that the enzyme alone is sufficient for complete degradation of 30S subunits. Collectively, our results provide a mechanistic basis for the degradation of 30S mediated by RNase R, and reveal that RNase R targets orphaned 30S subunits using a dynamic mechanism involving an anchored switching of binding sites.

Ribosomes are one of the most abundant machineries in the cell and are indispensable for growth. The number of ribosomes is highly regulated, being tightly coupled to the growth rate[2,3] and modulated by environmental conditions, such as the availability of nutrients[1,6–8]. Consequently, faulty ribosomes have to be removed from the translational pool to maintain high translational fidelity and to free up cellular resources during nutrient deprivation[4,5,9,10]. One of the most prominent exonucleases involved in the quality control and starvation-induced turnover of ribosomes in bacteria is the 3′ to 5′ exonuclease RNase R[4,11–14].

RNase R belongs to the RNB/RNase II family of enzymes and is homologous to yeast Rrp44 (DIS3 in humans), which forms the catalytic unit of the eukaryotic exosome[15,16]. Like other family members, *Bacillus subtilis* RNase R has a central RNB catalytic domain, flanked by two cold-shock domains (CSD1 and CSD2) at the N terminus, and an S1 domain with a lysine and arginine-rich tail (hereafter, K/R-rich tail) at the C terminus (Extended Data Fig. 1a). In addition, RNase R contains a unique helix-turn-helix (HTH) domain proximal to CSD1, which is absent in other family members, such as RNase II (Extended Data Fig. 1a). Despite having similar domain organizations, RNase II hydrolyses single-stranded RNA (ssRNA) substrates, whereas RNase R shows a preference for structured substrates that bear short ssRNA 3′ overhangs[17]. The catalytic RNB domain of RNase R is structurally similar to that of RNase II and Rrp44, consisting of a central channel with a lumen that can accommodate only ssRNA substrates[18,19] (Extended Data Fig. 1b). Thus, before entering the catalytic pocket of RNase R, RNA duplexes are thought to be unwound by the concerted action of the CSD and S1 domains that encircle the entry to the lumen of the RNB domain[18,19]. Although ATP is not required to unwind its substrates, RNase R is still extremely efficient at degrading highly structured RNAs when compared with other exonucleases[20].

In *Escherichia coli*, co-deletion of RNase R (or RNase II) with polynucleotide phosphorylase (PNPase) is lethal[21,22], whereas deletion of RNase R and RNase II combined with a temperature-sensitive PNPase mutation causes an accumulation of truncated rRNA products[7,11,23,24]. This suggests that these exonucleases are involved in rRNA degradation through a mechanism involving initial endonucleolytic cleavages that produce accessible 3′ ends for subsequent exonucleolytic degradation[7,11,23,24]. The association of RNase R with 30S subunits and 70S ribosomes has been previously reported[25,26]. Moreover, it has also been shown that, in strains lacking hibernation-promoting factors (HPFs), the 16S rRNA undergoes extensive degradation in stationary phase in a process that depends on RNase R[25,27]. In addition, in vitro degradation assays have revealed that *Staphylococcus aureus* RNase R preferentially degrades 30S over 50S subunits, especially when isolated from strains that lack HPFs[25]. However, the mechanism by which RNase R recognizes and degrades such large ribonucleoprotein particles has so far remained unclear.

[1]Institute for Biochemistry and Molecular Biology, University of Hamburg, Hamburg, Germany. [2]Institute of Technology, University of Tartu, Tartu, Estonia. [3]Dubochet Center for Imaging (DCI) at EPFL, EPFL SB IPHYS DCI, Lausanne, Switzerland. [4]Centre for Engineering Biology (SynthSys), University of Edinburgh, Edinburgh, UK. ✉e-mail: Daniel.Wilson@uni-hamburg.de; Helge.Paternoga@uni-hamburg.de

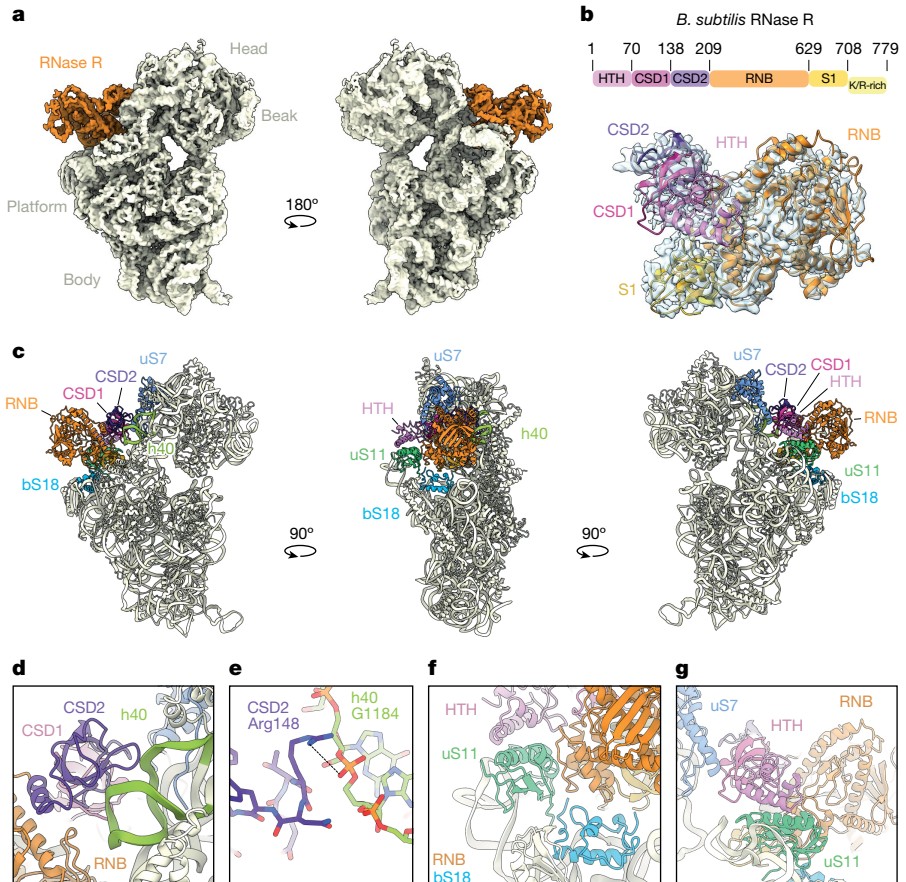

**Fig. 1 | Interaction of RNase R with the 30S subunit. a**, Two views of the unsegmented cryo-EM map of state I of the RNase R–30S complex, with the density for RNase R (orange) and 30S (pale yellow) coloured. The density has been filtered for visual clarity. **b**, Cryo-EM map density (transparent) and molecular model for RNase R, with RNase R domains coloured according to the schematic (HTH domain, pink; CSD1, magenta; CSD2, purple; RNase II family 3′ exonuclease domain (RNB), orange; S1 domain, gold; and K/R-rich tail, yellow). **c**, Overviews of the RNase R-binding site on the 30S subunit, highlighting interactions with 30S components h40 (green), uS7 (blue), bS18 (sky blue) and uS11 (lime). For RNase R, individual domains are indicated and coloured as in **b**. **d**, Interaction of the CSD2 domain (purple) with h40 (green). **e**, Interaction of Arg148 of CSD2 with G1184 in h40. **f**, Interaction of the HTH domain (pink) and the RNB domain (orange) with uS11 (lime) and bS18 (sky blue). **g**, Interaction of the HTH domain (pink) and the RNB domain (orange) with uS11 (lime) and uS7 (blue).

## Structure of RNase R on the 30S subunit

To provide insight into how RNase R mediates the degradation of ribosomal particles, we isolated native RNase R-ribosome complexes from *B. subtilis* grown to late-exponential phase. To this end, RNase R was C-terminally Flag-tagged and immunoprecipitated (Supplementary Fig. 2a,b), as performed previously for ribosome quality control factors[28,29]. However, the low cellular concentration of RNase R expressed from the endogenous locus precluded structural analysis, so a plasmid-based system was used to enhance expression (Supplementary Fig. 2c,d). Although overexpression of RNase R potentially generates non-specific interactions, a side-by-side comparison with the endogenous protein revealed near-identical banding patterns on RNA gels (Supplementary Fig. 2d). In addition to RNase R, co-immunoprecipitation of ribosomal proteins was observed (Supplementary Fig. 2a), which was confirmed by mass spectrometry (Supplementary Data). This result is consistent with the previous reports of RNase R interacting with ribosomal particles[25,26]. The native RNase R complexes were then subjected to single-particle cryo-electron microscopy (cryo-EM) analysis. Although two-dimensional (2D) classification indicated that most particles corresponded to 30S subunits (Extended Data Fig. 2), three-dimensional (3D) classification revealed that the 30S subunits exhibited high flexibility in the head region, which hampered the visualization of any bound factors, such as RNase R. Nevertheless,

after 3D classification of around 1.6 million starting ribosomal particles, we managed to obtain an initial class containing 89,890 particles (around 6% of the starting population) that exhibited additional density, which could be unambiguously attributed to RNase R. This class, in turn, gave rise to four subclasses, which differed in the position of the 30S head and the RNase R protein relative to the 30S body, consistent with the dynamic nature of the complex (Extended Data Fig. 2). All subclasses were refined further, yielding cryo-EM reconstructions of RNase R–30S complexes with average resolutions ranging from 3.1 to 4.2 Å (Extended Data Fig. 3a–h and Extended Data Table 1). The best resolved RNase R–30S complex, which we refer to as state I, had the highest particle number (28,143; 2%) and an average resolution of 3.1 Å (Fig. 1a and Extended Data Fig. 3a,b). Local resolution calculations indicated that RNase R is better resolved in the regions in which the factor interacts with the ribosome (around 3 Å), whereas the peripheral parts are more flexible and less well-resolved (around 5 Å) (Extended Data Fig. 3i–k). The other three subclasses, states I.1–I.3, are similar to state I, but with a shifted position of the 30S head (by up to 19.1 Å) and RNase R (by 4.8 Å) (Extended Data Fig. 3l–n and Supplementary Video 1).

In state I, RNase R is bound between the head and the body of the 30S subunit, adjacent to the exit site of the mRNA channel (Fig. 1a). With the exception of the C-terminal K/R-rich tail, the density for RNase R was sufficient to unambiguously assign all domains of the protein (Fig. 1b). The overall conformation of RNase R (including the CSD1,

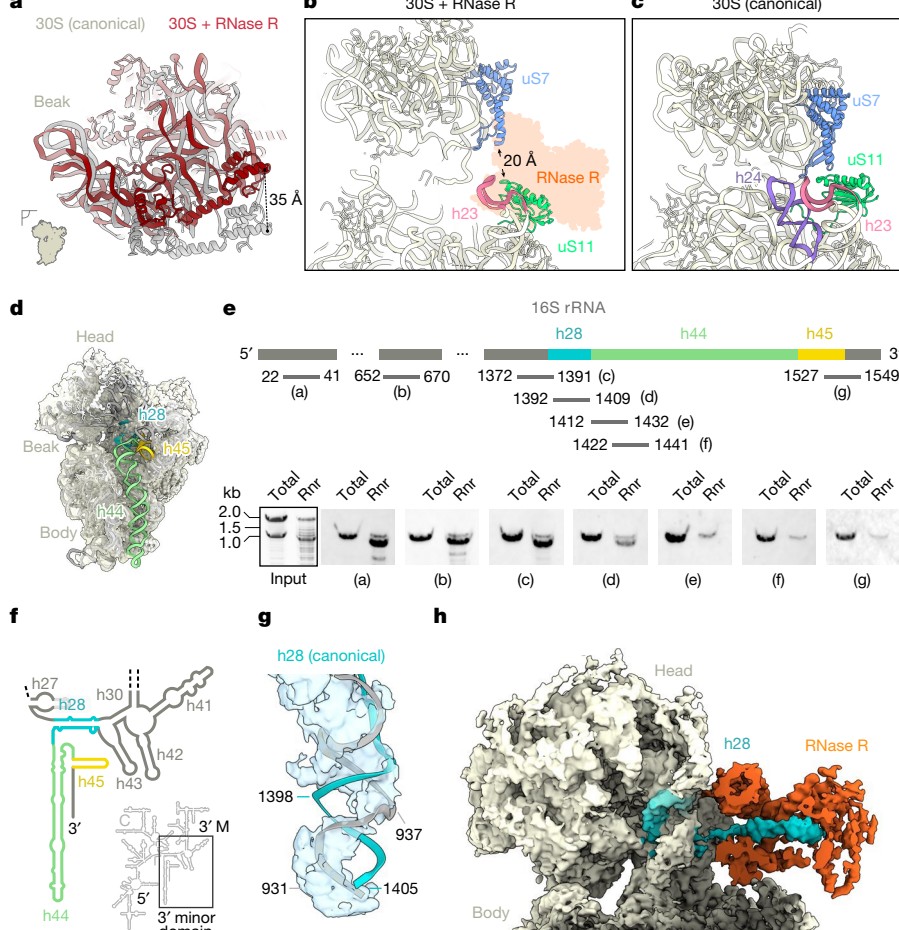

**Fig. 2 | RNase R is associated with a 30S degradation intermediate.**
**a**, Comparison of the 30S head position between a canonical state (grey, Protein
Data Bank (PDB) ID: 6HA8; ref. 33) and an RNase R-bound state I (dark red), when
aligned on the 30S body. **b,c**, Relative positions of uS7 (blue) and uS11 (lime)
with (**b**) and without (**c**) the presence of RNase R (PDB ID: 6HA8; ref. 33). Helix
h24 (purple) is disordered and not modelled in the RNase R-bound state in **c**.
**d**, Cryo-EM map of the RNase R-bound 30S fitted with the mature *B. subtilis* 30S
model (PDB ID: 6HA8; ref. 33), showing a lack of density for h28 (turquoise),
h44 (green) and h45 (yellow). The density has been filtered for visual clarity.
**e**, Northern blot analysis of immunoprecipitated RNA with probes (designated

by letter labels) as indicated in the schematic (*n* = 2). The schematic is not to
scale. The input was stained with Serva stain G. Rnr, RNase R immunoprecipitated
sample. **f**, Schematic of the secondary structure of *B. subtilis* 16S rRNA,
highlighting the 3′ minor domain that contains h28 (turquoise), h44 (green)
and h45 (yellow). 5′, 5′ domain; C, central domain, 3′ M, 3′ major domain.
**g**, Cryo-EM map of the neck region of 30S (state I) with a fitted model of the
canonical 16S rRNA (PDB ID: 6HA8; ref. 33). The density has been filtered for
visual clarity. **h**, Cryo-EM map with an isolated density for RNase R (orange) and
the h28 substrate (turquoise). The density has been filtered for visual clarity.
For gel source data, see Supplementary Fig. 1.

CSD2, RNB and S1 domains) observed here bound to the 30S is notably
similar to that reported previously for the ribosome-free structures
of *E. coli* and *Mycoplasma genitalium* RNase R[18,19], and also shares
similarities—albeit to a lesser extent—with other RNase II family exo-
nucleases, such as *E. coli* RNase II[30], yeast Rpr44 (ref. 16) and human
DIS3L2 (ref. 31) (Extended Data Fig. 1b). RNase R establishes interactions
with the head and body of the 30S subunit, predominantly using domains
located in the N-terminal portion of the molecule (Fig. 1c). Specifically,
the CSD1 and CSD2 domains of RNase R contact the 30S head, forming
interactions with ribosomal protein uS7 and 16S rRNA helix 40 (h40),
respectively (Fig. 1d). The latter interaction is well-resolved, such that
the side chain of Arg148 in CSD2 is observed to come within hydrogen-
bonding distance of the backbone of G1184 in h40 (Fig. 1e). The N-terminal
HTH domain and the tri-helix region (residues 477–534) in the RNB
domain of RNase R contact the platform region of the 30S body (Fig. 1c).
The tri-helix region of the RNB domain of RNase R is well-resolved and
inserts into a cleft between ribosomal proteins uS11 and bS18 (Fig. 1f).
Although the interaction with bS18 is stable and well-ordered, the entire
uS11 protein and the uS11-binding site in h23 appear flexible and poorly
ordered. The flexibility of uS11 is likely to explain why the N-terminal

HTH of RNase R is also poorly ordered, because it comes into close
proximity with uS11 (Fig. 1g). The C-terminal S1 domain of RNase R is
located deep within the cleft between the head and the body of the 30S
subunit, overlapping—but distinct from—the position observed for
the structurally related ribosomal protein bS1 (ref. 32) (Extended Data
Fig. 4a–c). No interaction between the S1 domain of RNase R and the 30S
subunit is apparent. Indeed, with the exception of the tri-helix region,
the RNB domain of RNase R also does not appear to make any addi-
tional interactions with the 30S, consistent with the dynamic motions
observed in the different RNase R subpopulations. The binding site of
RNase R overlaps with that of ribosomal protein bS21 (Extended Data
Fig. 4d–f), which is completely absent in state I, suggesting either that
RNase R displaces bS21 from the ribosome upon binding, or that dis-
sociation of bS21 is a prerequisite for RNase R binding.

## An RNase R–30S degradation intermediate

Further comparison of the RNase R–30S complex with the 30S subunit
from a *B. subtilis* 70S ribosome[33] revealed a number of conformational
changes that occur after the binding of RNase R (Fig. 2a, Extended

Data Fig. 3l–n and Supplementary Video 1). The largest movement is observed for the 30S head, which is tilted away from the intersubunit interface, leading to a shift of more than 35 Å at its periphery (Fig. 2a). By contrast, h23, including the associated ribosomal protein uS11, is shifted towards the intersubunit space, whereas the upper region of the neighbouring h24 has become completely disordered (Fig. 2b,c). Normally, uS7 and uS11 form a connection between the 30S head and body (Fig. 2c); however, the conformational changes observed in the presence of RNase R break this connection, leading to a separation of more than 20 Å between the two proteins (Fig. 2b). In addition to conformational changes, comparison of the RNase R–30S complex with the 30S from the *B. subtilis* 70S ribosome[33] revealed the absence of density for the 3′ end of the 16S rRNA, including helices h44 and h45, as well as part of h28 (Fig. 2d). One plausible explanation for this is that these regions are present, but not visualized in the cryo-EM map owing to extreme flexibility, as observed previously for some precursor 30S particles[34–38]. Alternatively, these regions might actually be absent owing to degradation of the 3′ end of the 16S rRNA by RNase R. To distinguish between these two possibilities, we isolated and analysed the rRNA species that co-immunoprecipitated with the Flag-tagged RNase R (Supplementary Fig. 2b,d). Unlike the lysate control with full-length 16S and 23S rRNAs, the immunoprecipitated RNase R–30S complex contained one major 16S rRNA species that is shorter by around 150 nucleotides (Fig. 2e and Supplementary Fig. 2b,d). To better define the truncation site, we used northern blotting with specific probes complementary to various regions of the 16S rRNA (Fig. 2e). This analysis confirmed that the 5′ end of the 16S rRNA is intact in the RNase R–30S complex (Fig. 2e, probe 'a'), whereas the 3′ end is truncated (Fig. 2e, probe 'g'). Moreover, the truncation site could be mapped to the vicinity of nucleotides 1392–1409 (Fig. 2e, probe 'd'), which are located within h28 that forms the 'neck' region connecting h44 to the 30S head (Fig. 2f). Collectively, this suggests that the RNase R–30S complex represents a 30S degradation intermediate that lacks around 150 nucleotides from the 3′ end, presumably owing to processive RNase R 3′ to 5′ exonuclease activity.

In canonical 30S subunits, helix 28 comprises nucleotides 932–945, which run from the 30S body towards the head, and form a 14-base-pair duplex with nucleotides 1387–1405 that, after folding of the head domain, return to the body to form h44 and h45. Careful inspection of the cryo-EM density for h28 in the RNase R–30S complex revealed that whereas the distal end of h28 (938–945/1398–1387) is base-paired, nucleotides 931–937 at the proximal end of h28 are single stranded, with the density lacking for their base-pairing partners 1399–1405 (Fig. 2g). Instead, we observed additional density for the missing 3′ nucleotides that passes behind the single-stranded region of h28 and extends towards the lumen of RNase R (Fig. 2h). Extra density is also observed within the lumen of RNase R, consistent with ssRNA as positioned within the *M. genitalium* homologue (Fig. 2h and Extended Data Fig. 1b). However, the density for the 3′ end of the 16S rRNA is not well-resolved and does not allow the sequence of the substrate to be unambiguously assigned, therefore we tentatively modelled a polyadenine sequence to illustrate the path. Collectively, our structure supports the use of the apical groove as entry site for the ssRNA substrate[18,19] and shows that we have captured RNase R in a state where it engages truncated 16S rRNA at the neck-to-head transition. In addition, this state appears to represent a roadblock, because to continue degradation of the 16S rRNA, the enzyme would have to thread the remaining 16S rRNA around neck nucleotides 931–945 (Fig. 2h).

## RNase R degrades 16S rRNA in mature 30S

The cryo-EM structure of the RNase R–30S complex suggests either that RNase R recognizes 30S subunits that lack the 3′ minor domain of 16S rRNA, or that it binds to mature 30S subunits to degrade the 3′ minor domain itself. Therefore, we used an in vitro degradation assay to work out whether *B. subtilis* RNase R can engage mature 30S subunits without the assistance of other nucleases, and to analyse possible degradation intermediates that result from the reaction. To do this, we recombinantly expressed and purified wild-type *B. subtilis* RNase R (Rnr^WT) as well as a *B. subtilis* RNase R(Asp260Asn) mutant (Rnr^D260N) (Supplementary Fig. 3a,b), equivalent to *E. coli* Rnr^D280N and *S. aureus* Rnr^D271N, which were previously shown to be catalytically inactive[19,25]. Consistent with its known capacity to degrade ssRNA substrates[25,39], we found that Rnr^WT, but not the Rnr^D260N mutant, could rapidly degrade a short linear ssRNA substrate (Supplementary Fig. 3c). In addition, Rnr^WT, but not the Rnr^D260N mutant, showed potent degradation activity against phenol-extracted forms of both the 16S and the 23S rRNAs (Fig. 3a), supporting previous observations that RNase R can also degrade duplex RNA[25,39]. We next assessed whether RNase R could degrade 16S and/or 23S rRNAs within the context of mature ribosomal subunits. To do this, we incubated Rnr^WT, or Rnr^D260N, with either 30S or 50S subunits and then analysed the remaining rRNA on denaturing gels (Fig. 3b,c and Supplementary Fig. 3d,e). The results clearly showed that Rnr^WT can efficiently degrade the 16S rRNA within the context of the mature 30S (Fig. 3b), but not the 23S rRNA within the 50S subunit (Fig. 3c), similar to that reported for *S. aureus* RNase R[25]. Notably, the degradation of the 16S rRNA from mature 30S appeared to proceed through a major degradation intermediate (Fig. 3b), which corresponded in size to the intermediate detected in our in vivo pull-downs (Fig. 2e and Supplementary Fig. 2b,d). To compare the truncation sites between the in vitro assay and the in vivo pull-outs, we sequenced the corresponding rRNAs in high throughput. The results revealed a major truncation site at nucleotide C1391 in both the in vitro and the in vivo samples (Extended Data Fig. 5). In addition, the in vivo samples contained extra sites extending towards C1412, with a minor peak at U1402 (Extended Data Fig. 5). These findings are consistent with our northern blot analysis of the in vivo sample, in which we observed a reduced signal for probe d, covering nucleotides 1392–1409, as well as a complete loss of signal for probe e, covering nucleotides 1412–1432 (Fig. 2e). We conclude that the major 16S rRNA degradation intermediate observed in our in vivo and in vitro experiments is a direct product of RNase R activity and does not depend on the presence of other RNases.

The current model of RNase R action is that the rRNA is endonucleolytically cleaved before becoming a substrate for RNase R[23,24]. Our data, however, indicate that RNase R itself is capable of not only initiating, but also fully degrading the 16S rRNA in the context of a mature 30S subunit (Fig. 3b). We note that this does not exclude the additional contribution of other endonucleases to facilitate the degradation process in vivo; however, it does raise the question of how mature 30S subunits protect themselves from RNase R action in the actively growing cell. Here, we considered two alternative scenarios. In the first scenario, we postulated that the presence of mRNA protects the 30S from degradation by RNase R during translation initiation. Because RNase R is an enzyme that accepts duplex RNA as a substrate, but needs a short single-stranded sequence to start[40,41], we thought that interaction between the Shine–Dalgarno (SD) sequence of mRNA and the anti-SD sequence located in the 3′ end of the 16S rRNA might block RNase R from starting the degradation reaction. To test this, we performed an in vitro 30S degradation assay in the presence of Rnr^WT and a short oligonucleotide complementary to the anti-SD region (Fig. 3d). In the presence of the SD oligonucleotide, we observed increased protection of the 16S rRNA against Rnr^WT-mediated degradation, as compared with a scrambled control oligonucleotide (Fig. 3d). This observation suggests that the presence of mRNA during initiation should provide protection against RNase R-mediated degradation.

In the second scenario, we hypothesized that the 16S rRNA is also protected from RNase degradation in the context of a 70S ribosome. Our rationale was that the binding of RNase R induces conformational changes within h23 and h24 that are at the subunit interface (Fig. 2a–c), and which may not be possible in the context of a 70S ribosome. Moreover, in a 70S ribosome, h44 and h45, which are located within the 3′ end

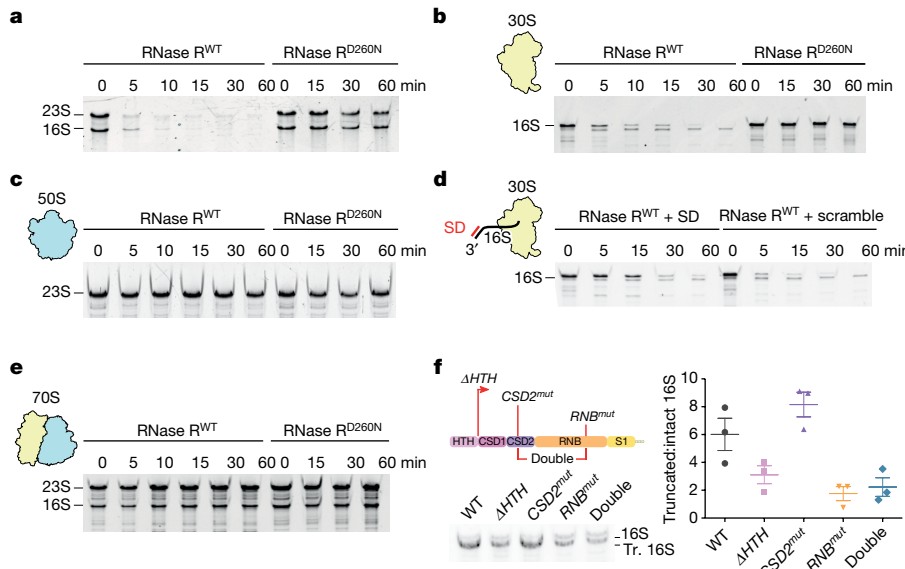

**Fig. 3 | RNase R degradation of 16S rRNA within mature 30S subunits.**
**a**–**c**, In vitro degradation assays of isolated total RNA (**a**; $n = 3$) and 30S (**b**; $n = 3$) and 50S (**c**; two technical replicates were performed) subunits, catalysed by recombinantly purified wild-type RNase R ($Rnr^{WT}$) and the catalytically inactive RNase R mutant ($Rnr^{D260N}$). RNase R proteins were mixed with the substrate and incubated at 37 °C for 0–60 min, after which the RNA was extracted and analysed on 6% denaturing TBE-Urea gels. **d**, In vitro degradation assay of isolated 30S subunits in the presence of DNA oligo (SD) which contains an SD sequence and is a reverse complement to the 3′ end of 16S rRNA (lanes 1–5). The control is a scrambled version of the SD oligo (scramble) (lanes 6–10). The ribosomes were pre-incubated with the oligos for 5 min at 37 °C before addition of the enzyme (two technical replicates were performed). **e**, In vitro degradation assay of isolated 70S ribosomes performed as in **b**–**d** ($n = 2$). **f**, Northern blot analysis of immunoprecipitated RNA from RNase R wild type (WT), $\Delta HTH$ mutant (in which the first 70 amino acids of the protein are deleted), $CSD2$ mutant (ETRN147GSGS), $RNB$ mutant (DRP518AAA) and a double $CSD2/RNB$ mutant. The ratios of truncated (Tr.) and intact 16S rRNA are plotted. Data are mean ± s.e.m. A two-tailed $t$-test showed significance ($P < 0.05$) between the WT and $\Delta HTH$ ($P = 0.032$), WT and $RNB$ ($P = 0.018$) and WT and $CSD2/RNB$ ($P = 0.02$) mutants, and no significance between the WT and $CSD2$ mutants ($P = 0.672$). Data are from $n = 3$ biologically independent experiments. For gel source data, see Supplementary Fig. 1.

---

of the 16S rRNA, establish multiple intersubunit bridges with the 50S subunit that might also hamper the action of RNase R. To assess this scenario, we performed the in vitro degradation assay using purified *B. subtilis* 70S ribosomes in the presence of 15 mM $MgCl_2$ to ensure that the subunits were tightly associated (Fig. 3e and Supplementary Fig. 3f). Under these conditions, we observed no obvious degradation of the 16S rRNA (Fig. 3e), which suggests that these 70S ribosomes are refractory to the action of RNase R. To ensure that RNase R is still active under these higher concentrations of $Mg^{2+}$, we also performed the in vitro degradation assay with isolated 30S at 15 mM $MgCl_2$, and found that RNase R remains active under these conditions, although it is less efficient (Supplementary Fig. 3g). Of note, comparison of the in vitro 30S degradation assay at lower (Fig. 3b) and higher $MgCl_2$ concentrations revealed that extra degradation intermediates were present at the higher $Mg^{2+}$ concentration (Supplementary Fig. 3g), suggesting that stabilization of the 16S rRNA secondary structure hampers the action of RNase R.

To understand how individual RNase R domains contribute to the 30S turnover reaction, we designed three variants based on our structural model (Fig. 1d–g): a truncation of the HTH domain ($\Delta HTH$; RNase R(71–779)); a mutation of the CSD2 linker that interacts with h40 ($CSD2^{mut}$: ETRN147GSGS); and a mutation of a short stretch in the RNB domain that interacts with uS18 ($RNB^{mut}$: DRP518AAA) (Fig. 1d–g). Subsequently, we performed a northern blot analysis of RNA co-purified by these RNase R variants (Fig. 3f). This analysis showed that the $\Delta HTH$ and $RNB^{mut}$ mutants and a $CSD2^{mut}/RNB^{mut}$ double mutant—but not the $CSD2^{mut}$ alone—purify reduced amounts of the truncated 16S rRNA species (Fig. 3f). These findings suggest that the HTH and RNB domains are important for the association of RNase R to 30S subunits during the initial degradation of the 3′ minor domain (Fig. 3f).

Finally, to understand whether there is a specific effect of RNase R on 30S degradation in vivo, we assessed the 30S content in $\Delta rnr$ cells compared with the wild-type strain. To do this, we analysed ribosomal profiles from cells grown to late-exponential phase, matching the condition of our pull-outs (Extended Data Fig. 6). Here, we observed an increased 30S peak in $\Delta rnr$ cells, with similar levels of 50S and a minor reduction in 70S ribosomes (Extended Data Fig. 6a), suggesting an accumulation of free 30S over 50S subunits in the $\Delta rnr$ strain. The 30S accumulation could result from reduced 30S turnover owing to the absence of RNase R. It might also arise if RNase R functions in the processing of ribosomal precursors, which cannot mature and therefore do not enter into the pool of translating 70S ribosomes. To test for the possibility that the larger 30S peak corresponds to biogenesis intermediates with immature 16S rRNA, we extracted RNA from serially collected sucrose gradient fractions (Extended Data Fig. 6b), and probed for 3′-extended 16S rRNAs using the $\Delta yqeH$ strain as a positive control for pre-16S accumulation[42]. This analysis revealed that neither the wild-type nor the $\Delta rnr$ samples contained measurable amounts of 16S rRNA with premature 3′ ends, whereas the expected signal[42] could be readily detected in the $\Delta yqeH$ strain (Extended Data Fig. 6c). These results suggest that the 30S accumulation we observe is not driven by early or intermediate 30S assembly intermediates. Nevertheless, we cannot exclude that very late biogenesis intermediates with mature 16S rRNA could contribute to the accumulation of 30S observed in the absence of RNase R.

## RNase R induces 30S head rearrangements

Our in vitro assays suggested that RNase R is able to fully degrade the 30S subunit, but our cryo-EM analysis of state I revealed that the enzyme encounters a kinetic barrier when transitioning from the neck to the head region of the 30S (Fig. 4a). We therefore sought to identify other states in our cryo-EM data that might reflect subsequent steps in the degradation process. After further in silico sorting, we were able to

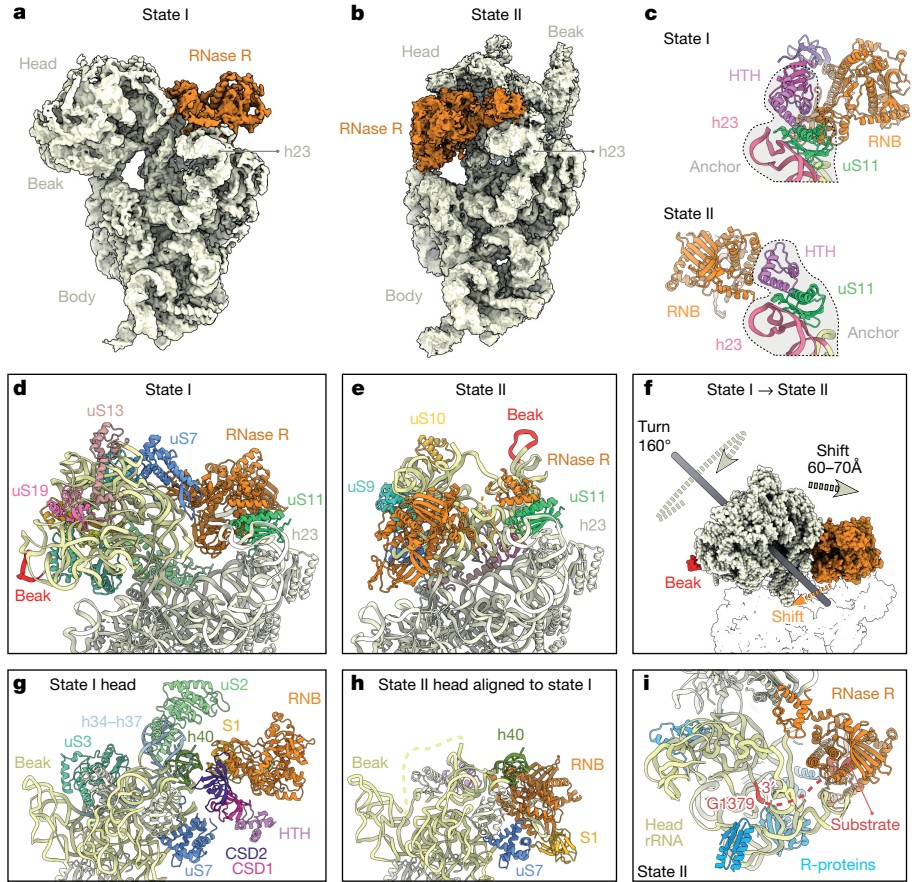

**Fig. 4 | RNase R-mediated 30S subunit degradation involves a major head rearrangement. a,b**, Comparison of the cryo-EM maps of RNase R (orange) in the 30S degradation state I (**a**) and in degradation state II (**b**). In both states the body of the ribosome has in the same orientation for reference. **c**, Comparison of the binding position of RNase R in state I (top) and in state II (bottom), with h24 (pink) and uS11 (green) for reference. **d,e**, Comparison of the head in state I (**d**) and state II (**e**), with the head rRNA (pale yellow) and the beak (red) coloured. **f**, Imaginary axes around which the 30S head and RNase R rotate to interconvert between states I and II. **g,h**, Relative binding positions of RNase R in state I (**g**) and state II (**h**), illustrating the shift in the RNB (orange) and S1 (yellow) domains. **i**, Proximity of RNase R (orange) to the 3′ end of the 16 rRNA (red) in state II, with neighbouring ribosomal proteins (R-proteins) (blue) and head rRNA (yellow) coloured.

identify a second stable state, which we refer to as state II, containing 4,011 particles (around 0.3% of all particles) (Extended Data Fig. 2). Despite the low number of particles, we were able to refine the sub-population to an average resolution of 4.7 Å (Extended Data Fig. 7a,b), which was sufficient to distinguish and assign the RNase R and the 30S subunit densities (Fig. 4b). In state II, RNase R has shifted from its location near the mRNA exit channel to the intersubunit surface, where it sits between the head and the body of the 30S (Fig. 4a,b). Although the density for RNase R in state II appears to be highly mobile, we could satisfactorily fit the HTH, RNB and S1 domains in the isolated cryo-EM map density (Extended Data Fig. 7c). Juxtaposition of RNase R in the initial state I with the rearranged state II revealed that the movement of the enzyme is facilitated by the HTH domain, which remains close to its initial position in state I and therefore probably serves as an anchor through its contacts with uS11 (Fig. 4c and Supplementary Video 2). Furthermore, density for the CSD1 and CSD2 domains is highly fragmented in state II. This suggests that these domains are highly flexible in this state, potentially engaging and destabilizing the flexible rRNA of the neck region.

In addition to RNase R, the entire 30S head has undergone a marked rearrangement in state II (Fig. 4d–f). When compared with state I, the head in state II is rotated by 160° and then further shifted by 60–70 Å so that the 30S beak is now positioned above the body platform (Fig. 4d–f and Supplementary Video 3). A head rotation of 160° observed here is unprecedented, because a maximum head swivel of 22° is possible

during canonical translocation events[43]. Moreover, the densities for ribosomal proteins uS2 and uS3, which normally bridge the 30S head and body (Fig. 4g), are completely absent in state II (Fig. 4h), presumably as a consequence of the head rearrangement. In state II, RNase R interacts exclusively with the head of the 30S subunit, forming interactions from the RNB domain with h40 and uS7 (Fig. 4h). Although RNase R also interacts with h40 and uS7 in state I, the binding and interaction mode of RNase R in state II is distinct from that observed in state I (Fig. 4g,h). Indeed, comparison of the binding position of RNase R between states I and II reveals a completely different orientation in state II (Fig. 4g,h and Supplementary Video 3). This suggests that if RNase R does maintain head interactions during the head rotation that transforms state I to state II, then the RNB domain of RNase II must subsequently disengage and then re-engage the 30S head at a different site. In state II, the RNB domain of RNase R occupies the position where the neck region (h28) was present in state I, and the neck region is completely disordered and/or degraded. As in state I, clear density for the rRNA substrate is observed within the RNB domain, which was modelled as a polyadenine sequence owing to the limited resolution (Fig. 4i). The substrate is fragmented beyond the substrate channel of RNase R, with the first resolved 3′ nucleotide of the 16S rRNA being G1379. We cannot therefore ascertain whether the enzyme has digested additional portions of the 16S rRNA compared to state I, but the missing densities for uS2 and uS3 and the head rearrangement suggest that state II follows state I within the timeline of RNase R-mediated

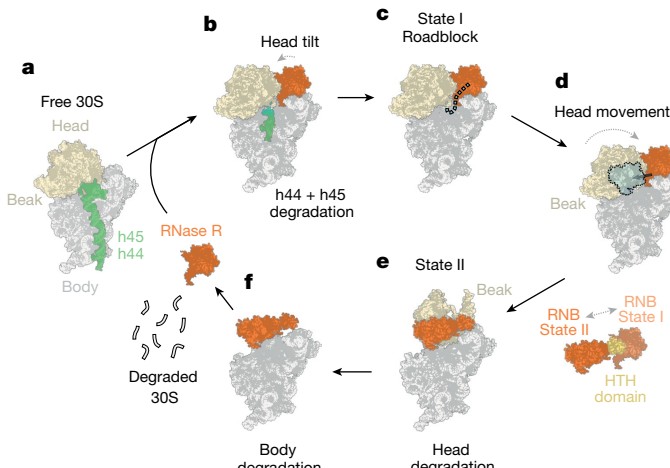

**Fig. 5 | Model of RNase R-mediated 30S subunit degradation. a**, RNase R is a processive enzyme and at first targets the single-stranded 3′ end of the 16S rRNA. **b**, Binding of RNase R leads to head movements that open the mRNA channel and facilitate the degradation of the highly structured h45 and h44. **c**, During the degradation of part of h28, RNase R reaches the 30S head, which poses a steric hindrance; this allows the first energetically stable degradation intermediate (state I) to be captured by cryo-EM. **d**, Because half of h28 is degraded, the neck region becomes highly flexible, leading to movement of the 30S head. **e**, Eventually, the 30S head is further destabilized, leading to a marked rearrangement in which uS2 and uS3 are displaced, the head rotates by 160° and RNase R has moved position, using the HTH as an anchor. This state corresponds to the second stable degradation intermediate (state II), which was also visualized by cryo-EM. The rearrangements allow RNase R to continue the degradation of the 30S head. **f**, Eventually, the complete 30S subunit can be accessed and degraded, and RNase R can dissociate and rebind another 30S subunit to initiate a further round of degradation.

30S subunit degradation. In state II, RNase R is now ideally positioned to attack the central region of the rRNA in the 30S head region, which would result in the destabilization of adjacent 30S ribosomal proteins owing to the removal of their rRNA substrate and subsequent unwinding of the entire head.

## Discussion

Together, our findings enable us to propose a model for RNase R-mediated degradation of the 30S subunit (Fig. 5a–f). First, our study reveals that RNase R binds initially to the 30S subunit at a site located between the 30S head and platform, where the mRNA exit site is located (Fig. 5a,b). Although our data indicate that mature 30S subunits are rapidly degraded by RNase R, we cannot exclude the possibility that late biogenesis intermediates with mature 16S rRNA are also substrates for degradation. After the initial binding, RNase R initiates the degradation of the 3′ end of the 16S rRNA, which encompasses the anti-SD sequence (Fig. 5b). We observe that the binding of RNase R induces conformational changes within the 30S subunit, involving a 20–30-Å movement of the 30S head away from the 30S platform, which widens the mRNA channel (Figs. 5b and 2b,c). We propose that this movement facilitates the degradation of structured regions of h44 and h45, because ultimately these rRNA elements are located on the intersubunit side and therefore need to be passed through this corridor to reach the lumen of RNase R (Fig. 5b). After the degradation of h44 and h45, RNase R proceeds with the degradation of one strand of h28, before encountering a roadblock to further degradation of the 30S head (Fig. 5c). The loss of the integrity of h28, which comprises the neck of the 30S head, leads to increased mobility of the 30S head relative to the body, as observed when comparing states I.1–3 (Fig. 5b and Supplementary Video 1). We suggest that this mobility enables

RNase R to eventually escape the roadblock by inducing a major conformational rearrangement in the position of the 30S head; namely, a 160° rotation and 60–70-Å shift, so that the beak of the 30S head is now located above the platform, as seen in state II (Fig. 5d,e and Supplementary Video 3). The observed head movement is accompanied by a relocation of the RNB domain of RNase R, moving from the 30S platform to the subunit interface, using the HTH domain of RNase R as an anchor (Fig. 5d,e and Supplementary Video 2). Comparison of the structures of states I and II suggests that RNase R disengages and re-engages the 30S head to enable continued degradation of the 16S rRNA that comprises the 30S head (Fig. 5e). Therefore, we propose that RNase R uses both processive and distributive (dissociation and rebinding) activities during the degradation of the 30S subunit, enabling it to overcome any potential roadblocks that it encounters. Although our data suggest that RNase R alone is sufficient to mediate the complete degradation of the 30S subunit (Fig. 5f), it is likely that other nucleases facilitate the process in vivo. Similarly, although we show that RNase R does not degrade 50S subunits in vitro (Fig. 3c), one could imagine that in vivo, the 50S subunit could become a substrate for RNase R through a preceding endonuclease cleavage. In *E. coli*, *B. subtilis* and *Streptomyces coelicolor*, the addition of 3′ overhangs through polyadenylation has been proposed to target defective rRNAs for degradation[44–47]; however, whether rRNAs become polyadenylated and degraded by RNase R in *B. subtilis* has not, to our knowledge, been examined.

The efficiency with which RNase R can degrade mature 30S subunits in vitro raises the question of how this process is regulated in the cell. Because RNase R initiates degradation by binding to the platform and accessing the single-stranded 3′ end of the 16S rRNA to initiate degradation, one can envisage that RNase R action can be blocked, and possibly even regulated, by factors or ligands that prevent access to this region of the ribosome. We rationalized that actively initiating and translating ribosomes might be refractory to the action of RNase R because the mRNA might block the access of RNase R to the 3′ end of the 16S rRNA. Indeed, we observe extensive overlap between an SD–anti-SD duplex that forms on the ribosome[48] and the binding site of RNase R (Extended Data Fig. 8a–c). Furthermore, we observe that SD oligonucleotides that are complementary to the 3′ end of the 16S rRNA interfere with RNase R-mediated 30S subunit degradation (Fig. 3d). These results are consistent with a previous report, which showed that 30S subunits are more prone to degradation than are 50S subunits in a cell-free translation system, and that active translation prevents ribosome degradation to some extent[49].

We note that during late stages of 30S assembly, when the RNase R-binding site is formed, the action of RNase R might also be prevented by the presence of biogenesis factors, many of which interact with this area of the 30S subunit[38], as seen for example for RbfA (ref. 50) (Extended Data Fig. 8d–f). Our structure of the RNase R–30S complex also rationalizes the observed protection from RNase R that is conferred by the HPF[25,27], because the binding site for RNase R observed in state I would overlap with the C-terminal domain of HPF and, in particular, with the linker that connects the N- and C-terminal domains (Extended Data Fig. 8g–i). Moreover, HPF-mediated formation of hibernating 100S ribosomes[51–54] would completely occlude the RNase R-binding site on the platform of the 30S subunit (Extended Data Fig. 8j,k).

In addition to stationary phase and starvation conditions, the targeted degradation of 30S subunits might also occur as part of ribosome-associated quality control (RQC) pathways[29]. In *B. subtilis*, collisions of bacterial ribosomes are sensed by the ATPase MutS2, which is proposed to promote the dissociation of ribosomal subunits[55]. Although the resulting 50S–peptidyl-tRNA complexes are subject to the action of RqcH and RqcP[55], the fate of the 30S subunit remains unclear. Further studies will be needed to address the potential role of RNase R in 30S degradation in the context of RQC, and particularly in scenarios in which ribosomal stalling results from damage to the 30S subunit.

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

# Methods

## Plasmid construction

Cloning was performed essentially as described[56]. Mutagenesis was conducted according to the Naismith protocol[57]. To clone the *B. subtilis rnr* (RNase R) gene, first the plasmid pHT01 p43 wRBS MCS-GS5-C-Flag tGyrA was created. To this end, the plasmid pHT01 p43 wRBS RsfS-Flag tGyrA[55] was digested by XbaI and NotI and ligated with annealed primers containing a multiple cloning site followed by an encoded 5×GS-linker and C-terminal Flag tag. The RNase R open reading frame (ORF) was PCR-amplified from *B. subtilis* genomic DNA (extracted from BGSC strain 1A1 wild type: trpC2) and inserted between the XbaI and BamHI sites of plasmid pHT01 p43 wRBS MCS-GS5-C-FLAG tGyrA to create plasmid pHT01 p43 wRBS RNase R-GS5-Flag tGyrA. For the expression of recombinant wild-type and mutant RNase R, PCR-amplified ORFs were inserted into the NdeI and BamHI sites of pET24d-His6-Tev[58].

## Strain preparation

To create the endogenously tagged RNase R–Flag strain, the wild-type strain (BGSC strain 1A1 wild type: trpC2) was transformed with the product of an overlap extension PCR encompassing a 5× GS-linker, the Flag tag and the chloramphenicol resistance marker *cat*, flanked by regions of around 1.2 kb homologous to the *rnr* locus. To avoid disturbing downstream gene expression, no additional promoter was introduced before the *cat* gene, resulting in expression only from the endogenous *rnr* promoter(s). The procedure involved several steps: first, a template vector was prepared. To this end, an inverse PCR was performed on the vector pHT01 (MoBiTec) using the primers 'iPCR_GS5-FLAG-CAT_fw' and 'iPCR_GS5-FLAG-CAT_rv' creating plasmid 'pHT01_INT_5xGS-C-Flag_cat' with the 5×GS-linker, Flag tag and *cat* marker in the correct order (Supplementary Table 1). Then three PCRs were performed: PCR[OVER1] using primers 'over1_fw' and 'over1_rv' on genomic DNA; PCR[OVER2] using primers 'over2_fw' and 'over2_rv' on plasmid 'pHT01_INT_5×GS-C-Flag_cat' and PCR[OVER3] using primers 'over3_fw' and 'over3_rv' on genomic DNA (Supplementary Table 1). PCR[OVER1] and PCR[OVER3] contain complementary regions to PCR[OVER2] and were thus combined in a single overlap extension PCR reaction. The product was purified from an agarose gel and directly transformed into the *B. subtilis* wild-type strain (BGSC strain 1A1 wild type: trpC2). Correct integration was confirmed by colony PCR using primers 'COL_fw' and 'COL_rv', as well as by sequencing of the resulting PCR products. The primer sequences are listed in Supplementary Table 2.

## Flag purification and RNA immunoprecipitations

*B. subtilis* 168 wild-type cells (BGSC strain 1A1 wild type: trpC2) expressing RNase R–Flag from pHT01 p43 wRBS RNase R-GS5-Flag tGyrA were grown at 37 °C in lysogeny broth (LB) medium (Roth) supplemented with 5 µg ml$^{-1}$ chloramphenicol and shaking at 145 rpm until an optical density at 600 nm ($OD_{600 nm}$) of 1.5. Cells were collected in 25 mM HEPES-KOH pH 7.5, 100 mM potassium acetate, 15 mM magnesium acetate, 0.1% NP-40 and 0.5 mM Tris carboxy ethyl phosphene (TCEP) buffer supplemented with protease inhibitor cocktail (Roche), flash-frozen in liquid nitrogen and lysed under cryogenic conditions using a Retsch MM400 (Retsch). The lysate was cleared at 16,000 rpm for 15 min and incubated with anti-Flag M2 affinity beads (Merck) for 1.5 h at 4 °C on a turning wheel. After in-batch wash with 20 ml lysis buffer without protease inhibitors, the beads were transferred to a Mobicol column and washed with 4 ml of 25 mM HEPES-KOH, pH 7.5, 100 mM potassium acetate, 15 mM magnesium acetate, 0.01% DDM (dodecylmaltoside) and 0.5 mM TCEP buffer, after which the RNase R complexes were eluted using 0.2 mg ml$^{-1}$ 3× Flag peptide (Sigma) in wash buffer. The complexes were then applied to grids for electron microscopy analysis or analysed on 4–12% NuPAGE SDS–PAGE gels (Invitrogen) by staining with Instant Blue (Expedeon). To extract RNA, 1 ml of Trizol reagent (Invitrogen) was added to the eluate and the

extraction was performed according to the manufacturer's instructions. The extracted RNA was then mixed with 2× RNA gel loading dye (Thermo Fisher Scientific), heated for 5 min at 65 °C and analysed on 6% TBE-Urea gels (Invitrogen). Subsequent analysis included either staining with Serva Hisense Stain G (Serva) or northern blotting (see below).

## Purification of recombinant RNase R (wild type and D260N mutant)

BL21-CodonPlus(DE3)-RIL cells (Stratagene) transformed with expression vectors were grown in LB medium (Roth) supplemented with antibiotics (30 µg ml$^{-1}$ kanamycin and 34 µg ml$^{-1}$ chloramphenicol). His6-TEV tagged RNase R wild type and the D260N mutant were expressed by induction with isopropyl β-D-1-thiogalactopyranoside (IPTG) for 16 h at 18 °C. After collection, the cells were lysed by a microfluidizer processor (Microfluidics) in 1 M NaCl, 20 mM Tris-HCl (pH 7.5), 5% glycerol, 0.01% NP-40 and 40 mM imidazole buffer, supplemented with protease inhibitor cocktail (Roche cOmplete EDTA free) and 0.5 mM TCEP. The lysates were subsequently cleared by centrifugation at 16,000 rpm for 20 min and applied to a HisTRAP Ni column (GE Healthcare). The bound proteins were eluted over an imidazole gradient (40–600 mM). After initial size-exclusion chromatography on a Superdex 200 16/600 column, the His6-tag was cleaved by acTEV protease (Thermo Fisher Scientific) and the proteins were passed one more time over a HisTRAP Ni column to remove the tag and protease. Finally, the flow-through was concentrated and the proteins were further purified by size exclusion on a Superdex 200 16/600 column equilibrated in 20 mM HEPES-K (pH 8), 150 mM KCl, 2 mM MgCl$_2$, 5% glycerol and 0.5 mM TCEP buffer.

## Isolation of ribosomes

Ribosomes were isolated essentially as described[59]. In brief, The *Δrnr* strain (BGSC BKE33610 trpC2; *Δrnr*::erm) was grown in 2 l LB medium until $OD_{600 nm}$ = 0.8. The cells were flash-frozen in ribosome buffer (20 mM HEPES-KOH, pH 7.5, 6 mM magnesium acetate, 30 mM NH$_4$Cl and 0.5 mM TCEP) and lysed under cryogenic conditions using a Retsch MM400 (Retsch). The lysate was pre-cleared at 17,000 rpm for 30 min at 4 °C. The supernatant was then centrifuged for 17 h at 40,000 rpm in a Beckman 70.1 Ti rotor to pellet 70S ribosomes and remaining polysomes. The crude ribosomes were resuspended by gentle shaking at 4 °C for 60 min in either ribosome or dissociation buffer (20 mM HEPES-KOH, pH 7.5, 1 mM magnesium acetate, 200 mM NH$_4$Cl and 0.5 mM TCEP) for subsequent isolation of 70S or ribosomal subunits, respectively. The resuspended ribosomes were subsequently loaded on 10–30% sucrose gradients and run in a Beckman SW32Ti rotor at 18,000 rpm for 19.5 h. The peaks corresponding to 30S, 50S and 70S were collected and the 70S ribosomes, or subunits, were pelleted further at 40,000 rpm for 22 h in a Beckman Ti70 rotor, or at 47,000 rpm for 20 h in a Beckman Ti70.1 rotor.

## Total RNA preparation

For the preparation of total RNA, 2 ml of wild-type culture (BGSC strain 1A1 wild type: trpC2) with $OD_{600 nm}$ = 1.8 was collected and the cells were resuspended in 1 ml Trizol reagent, after which the RNA was extracted according to the manufacturer's instructions.

## Linear RNA substrate preparation

A template encompassing a sequence upstream of the stalling loop of Erm BL (TAATACGACTCACTATAGGGAGACTTAAGTATAAGGAG GAAAAAATATGTTGGTATTCCAAATGCGTAATGTAGATAAAACATCTAC TATTTGAGTGATAGAATTCTATCGTTAATAAGCAAAATTCATTATAACC)[60] was PCR-amplified using an oligo containing the T7 promoter sequence. The PCR product was subsequently used as a template for in vitro transcription using the T7 MEGAscript kit (Invitrogen) according to the manufacturer's instructions.

## In vitro degradation assays

Final concentrations of 200 nM of RNase R wild type or mutant were mixed with linear RNA (600 nM), total extracted RNA (0.1 µg µl$^{-1}$), 30S (60 nM), 50S (40 nM) or 70S (40 nM) in 20 mM HEPES-K (pH 8), 150 mM KCl, 2 mM or 15 mM MgCl$_2$, 5% glycerol and 0.5 mM TCEP buffer and incubated at 37 °C for different times between 0 and 60 min. The RNA was extracted with Trizol (see above) and analysed on 6% TBE-Urea gels (Invitrogen). For the experiment with SD and scramble DNA oligos, the oligos (200 nM, final concentration) were mixed with the ribosomes and pre-incubated for 5 min at 37 °C before addition of RNase R.

## Northern blots

For northern blots, 600 ng of RNA extracted from the RNase R immunoprecipitation (see above) and 420 ng of total RNA were loaded on 6% TBE-Urea gel (Invitrogen). The gel was run for 1.5 h at 180 V in 1× TBE (Tris-Borate-EDTA) buffer (Thermo Fisher Scientific), after which the blot was conducted on an Amersham Hybond-N+ membrane (Cytiva) in a wet-blot transfer chamber (Bio-Rad) with 0.5× TBE buffer overnight at 40 V (4 °C). The membrane was then dried at 65 °C for 10 min and cross-linked in a Stratagene UV cross-linker (twice at automode). After blocking at 28 °C for 1.5 h in 250 mM Na$_2$HPO$_4$ pH 7.2, 1 mM EDTA, 7% SDS, 0.5% BSA (Applichem) and 80 µg ml$^{-1}$ salmon sperm DNA (Sigma) buffer, 0.5 pmols 5′-Cy3-labelled ssDNA probe (Metabion) was added and the membrane was incubated overnight on a turning device at 28 °C. After washing twice with 2× SSC buffer and 0.2% SDS, and twice with 1× SSC buffer and 0.1% SDS, the blot was visualized using an Amersham Typhoon scanner (GE, Cytiva). The sequences of all probes are listed in Supplementary Table 1.

## Western blots

Fifty millilitres of cell culture was grown to an OD$_{600 nm}$ of 1.4 in LB medium at 37 °C with shaking at 145 rpm. The cells were collected and lysed in 250 µl of 20 mM HEPES-Na pH 7.5, 100 mM NH$_4$Cl, 10 mM magnesium acetate and 0.5 mM TCEP with 0.1 mm Zirconia-glass beads (Carl Roth) using a FastPrep-24 (Millipore). Clarification was performed at 14,000 rpm and 4 °C for 10 min. The supernatants were transferred to a fresh tube and samples we normalized by measurement of the absorption at 260 nm. The samples were run on 4–12% NuPAGE gel (Invitrogen) and blotted using a Trans-blot Turbo transfer pack (Bio-Rad) on a Bio-Rad Trans-Blot Turbo machine for 7 min. The membrane was stained at first with Ponceau S, photographed and then blocked with 5% skimmed milk in TBS–Tween (0.1%) for 30 min. The membrane was then incubated overnight with monoclonal anti-Flag M2–HRP antibody (Sigma, A8592) diluted 1:2,000 in 5% skimmed milk/Tris-buffered saline with 0.1% Tween-20 (v/v) (TBST). After washing twice with 5% skimmed milk/TBST and once with TBST, the signal was developed with Clarity Western ECL substrate (Bio-Rad) and visualized using the Bio-Rad ChemiDoc Imaging system.

## Sucrose gradients

Twenty-five millilitres of wild-type (BGSC strain 1A1 wild type: trpC2) and isogenic *rnrΔ* cells (BGSC BKK33610 trpC2; *Δrnr*::kan) were grown in LB medium at 37 °C and 145 rpm, then collected at OD$_{600 nm}$ = 1.4 and lysed in 250 µl of 20 mM HEPES-Na pH 7.5, 100 mM NH$_4$Cl, 10 mM magnesium acetate and 0.5 mM TCEP with 0.1 mm Zirconia-glass beads (Carl Roth) using a FastPrep-24 (Millipore). Clarification was performed at 14,000 rpm and 4 °C for 10 min. The supernatants were transferred to a fresh tube and a volume corresponding to 10 optical density units (OD$_{260 nm}$) was layered on top of a 10–40% (w/v) linear sucrose gradient and spun for 18.5 h at 19,000 rpm in a SW40 Ti rotor (Beckman Coulter). The ribosome profiles were then measured using a gradient station (Biocomp). For the northern blot analysis of RNA extracted from sucrose gradient fractions, control total RNA of wild-type and *ΔqyeH* (BGSC BKE25670; trpC2, *ΔyqeH::erm)* cells was used. Cells were grown to OD$_{600 nm}$ = 1.4 in LB medium at 37 °C and 145 rpm.

## Cryo-EM grid preparation and data collection

Sample volumes of 3.5 µl (8 OD$_{260 nm}$ per ml) were applied to grids (Quantifoil, Cu, 300 mesh, R3/3 with 3 nm carbon) which had been freshly glow-discharged using a GloQube (Quorum Technologies) in negative charge mode at 25 mA for 90 s. Sample vitrification was performed using ethane or propane in a Vitrobot Mark IV (Thermo Fisher Scientific), the chamber was set to 4 °C and 100% relative humidity and blotting was done for 3 s with no drain or wait time. Data were collected in an automated manner using EPU v.3.0 on a cold-FEG fringe-free Titan Krios G4 (Thermo Fisher Scientific) transmission electron microscope operating at 300 kV. The camera was operated in electron counting mode and data were collected at a magnification of 96,000× with the nominal pixel size of 0.83 Å and a nominal defocus range of −0.4 to −0.9 µm. A total of 23,349 micrographs in EER format were collected with 5.31 s of exposure (corresponding to a total dose of 50 e per A$^2$ on the specimen). No statistical methods were used to predetermine the sample size. The sample size was selected on the basis of a three-day data collection, which was chosen to obtain a sufficient number of particles for data processing.

## Cryo-EM data processing

Processing was performed using RELION 3.1.3 (refs. 61,62). The pixel size for processing was adjusted to 0.8 Å from the nominal 0.83 Å during data collection owing to best correlation with published ribosome models at this pixel size. Movie frames were aligned with MotionCor2 (ref. 63) using 4×4 patches followed by CTF estimation of the resulting micrographs using CTFFIND4 (ref. 64) using power spectra from the MotionCor run. The CTF fits were used to remove outlier micrographs with estimated resolutions greater than 15 Å, which retained 21,667 micrographs. crYOLO 1.8.0b47 with its general model (gmodel_phosnet_202005_N63_c17.h5) was used for particle picking, which resulted in 2,303,673 particles[65,66]. These were extracted in a box size of 64 px at a pixel size of 4.8 Å and subjected to 2D classification.

After 2D classification, 1,604,042 particles resembling 30S subunits were selected and used for a first 3D auto-refinement to centre all particles for further refinement steps. An empty mature 30S subunit was used as reference, with the initial volume being generated from PDB ID 6HA8 (ref. 33). Afterwards, particles were extracted with re-centring from the previous Refine3D-job at a box size of 128 px and a pixel size of 2.4 Å. The particles were aligned into a 3D volume using the output of the initial Refine3D-job as a reference (re-scaled to the new box and pixel sizes). From these aligned particles, 3D classification was performed without further angular sampling. Particle sorting was performed according to Extended Data Fig. 2. Particles for final classes of state I, state I.1–3 and state II were re-extracted at a box size of 384 px with a pixel size of 0.8 Å and subjected to 3D auto-refinement. Particles for state I and state I.1–3 were further CTF-refined to correct for anisotropic magnification, trefoil and higher-order aberrations, defocus and astigmatism. Furthermore, particles for state I were subsequently subjected to Bayesian polishing followed by another round of CTF refinements. After these procedures, the final volumes were generated by 3D auto-refinement and postprocessing in RELION.

## Molecular model building

The initial model for the 30S subunit of state I was generated based on a published *B. subtilis* 70S structure (PDB ID: 6HA8; ref. 33). This model was updated in Coot using protein restraints generated by ProSmart from AlphaFold models for all 30S ribosomal proteins[67–72]. For RNase R, an AlphaFold model was used and rigid-body fitted into the density using ChimeraX (refs. 69,72–74). Afterwards, the model was manually adjusted in Coot (refs. 67,68). Model refinement was performed using REFMAC5 as implemented in Servalcat (ref. 75). Subsequently, models

for state I.1–3 and state II were derived by iterative adjustment from the state I model. Cryo-EM data collection, refinement and validation statistics for all models are listed in Extended Data Table 1. When shown in figures, the RNase R S1 domain in state II was included as a separate rigid-body-fitted entity; fitting was performed using ChimeraX (refs. 73,74).

## Mass spectrometry

Protein pellets were taken up in 7 M urea, 2 M thiourea, 100 mM ammonium bicarbonate and 20 mM methylamine buffer at 0.5 µg µl$^{-1}$. Samples were then reduced with 10 mM dithiothreitol by incubating for 1 h at 25 °C, followed by alkylation with 20 mM chloroacetamide for 1 h at room temperature in the dark. Proteins were pre-digested with 1:100 (enzyme:protein) Lys-C (Wako) protease for 2 h at 25 °C, followed by fivefold dilution with 100 mM ammonium bicarbonate and overnight digestion with 1:100 trypsin (Sigma Aldrich) at 25 °C. Digests were acidified by bringing trifluoroacetic acid (TFA) to 1% and desalted on in-house-made C18 StageTips. Final liquid chromatography–tandem mass spectrometry (LC–MS/MS)-ready samples were constituted in 0.5% TFA ready for injection.

Five hundred nanograms of peptides were injected into an Ultimate 3500 RSLCnano system (Dionex) using a 0.3 × 5-mm trap-column (5-µm C18 particles, Dionex) and an in-house packed (3-µm C18 particles, Dr Maisch) analytical 50 cm × 75-µm emitter column (New Objective). Both columns were operated at 45 °C. Peptides were eluted at 300 nl min$^{-1}$ with an 8-42% B 60-min gradient (buffer B: 80% acetonitrile + 0.1% formic acid, buffer A: 0.1% formic acid) to a Q Exactive HF (Thermo Fisher Scientific) mass spectrometer (MS) using a nano-electrospray source (spray voltage of 2.5 kV). The MS was operated with a top-12 data-dependent acquisition strategy. In brief, one 350–1,400 $m/z$ MS scan at a resolution setting of $R = 60,000$ at 200 $m/z$ was followed by higher-energy collisional dissociation fragmentation (normalized collision energy of 26) of the 12 most intense ions ($z$: +2 to +5) at $R = 30,000$ with 1.6 $m/z$ isolation windows. MS and MS/MS ion target values were 3e6 and 1e5 with 50-ms and 41-ms injection times, respectively. Peptide match was set to preferred and exclusion of isotopes turned on. Dynamic exclusion was limited to 30 s.

MS raw files were processed with the MaxQuant software package (v.2.1.4.0). Methionine oxidation and protein N-terminal acetylation were set as potential variable modifications, whereas cysteine carbamidomethylation was defined as a fixed modification. Identification was performed against the UniProt (https://www.uniprot.org/) *B. subtilis* reference proteome database using the tryptic digestion rule. Only protein identifications with at least two peptides of a length of at least six amino acids (with up to two missed cleavages) were accepted. The intensity-based absolute quantification (iBAQ) feature of MaxQuant was enabled. This normalizes protein intensities by the number of theoretically observable peptides and enables a rough intra-sample estimation of protein abundance. The peptide-spectrum match, peptide and protein false discovery rate were kept below 1% using a target-decoy approach. All other parameters were default.

## RNA sequencing

For sequencing of the in vivo samples, the Flag immunoprecipitation was performed as described above and the RNA was extracted with 1 ml of Trizol reagent (Invitrogen) according to the manufacturer's instructions with an additional 75% ethanol wash step at the end. The experiment was performed in a biological triplicate. To sequence the in vitro degradation reactions, 5 µM RNase R was mixed with 1.5 µM 30S subunits in a 10 µl volume at 37 °C and the reaction was stopped at 4 min with the addition of 1 ml Trizol. For the control samples, the Trizol was immediately added without any incubation. The further purification was performed according to the manufacturer's instructions with an additional 75% ethanol wash step at the end. The samples were then taken into a modified NextFlex small RNA seq v.4 protocol.

Inputs were standardized to 206 ng as measured by RNA Qubit HS. Samples had 3′ adapters ligated with an adapter dilution of 1:1, followed by an adapter inactivation step (steps A and B). Samples were cleaned up with Adapter Depletion Solution, beads and isopropanol following step E but with reagent volumes adjusted for the smaller reactions. Samples were resuspended in 12 µl water, and 11.2 µl was taken, added with 4 µl of 5× T4PNK buffer (NEB) and fragmented at 94 °C for 1 min. Four microlitres of 10 mM ATP and 0.8 µl of T4PNK were added, and samples were incubated at 37 °C for 30 min, followed by deactivation at 65 °C for 20 mins. Then samples were taken into the 5′ ligation step from the NextFlex protocol with adapters diluted 1:3 (step C), and the remainder of the protocol was followed as per the manufacturer's instructions. The positive control was amplified with 16 PCR cycles, whereas the RNase R samples were amplified with 25 PCR cycles. All samples were cleaned up individually with a 1.3× bead ratio, and checked on the bioanalyser. Samples were pooled equimolarly, and cleaned up once more with a 1× bead ratio. Samples were sequenced using the MiSeq 50 bp v.2 kit with the following read mode: 5-8-0-61. Samples were demultiplexed using bcl2fastq, adapter trimming was performed with cutadapt and sequences from Read 2 were taken forward into alignments using Novoalign (https://www.novocraft.com/; v.3.06). After generating the bam files, bedgraph files were generated using bedtools and visualized using the IGV genome browser.

## Figure preparation

Molecular graphics were prepared with UCSF ChimeraX (refs. 73,74). The 16S rRNA secondary structure schematic was generated using R2DT (https://rnacentral.org/r2dt) with template 'Bacillus subtilis rRNA 16S d.16.b.B.subtilis' and the 16S rRNA sequence of locus BSU_rRNA_4/rrnA-16S, obtained from *Subti*Wiki (http://subtiwiki.uni-goettingen.de/)[76,77]. Figures were arranged using ImageJ[78] and Inkscape (https://inkscape.org/).

## Reporting summary

Further information on research design is available in the Nature Portfolio Reporting Summary linked to this article.

## Data availability

Cryo-electron microscopy maps have been deposited at the Electron Microscopy Data Bank as follows: state I, EMD-16595; state I.1, EMD-16606; state I.2, EMD-16605; state I.3, EMD-16607; and state II, EMD-16596. Associated molecular models have been deposited at the PDB: state I, 8CDU; state I.1, 8CED; state I.2, 8CEC; state I.3, 8CEE; and state II, 8CDV. The sequencing data related to Extended Data Fig. 5 can be accessed at the NCBI Gene Expression Omnibus with the accession number GSE251701.

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

**Acknowledgements** Part of this work was performed at the Multi-User CryoEM Facility at the Centre for Structural Systems Biology, Hamburg, supported by the Universität Hamburg and Deutsche Forschungsgemeinschaft (DFG) grant numbers INST 152/772-1, 152/774-1, 152/775-1, 152/776-1 and 152/777-1 FUGG). We thank the EMBL GeneCore facility and particularly D. Welter, Z. Henseler and V. Benes for performing the RNA sequencing and initial sequence analysis; F. Cornejo, K. Turgay and C. Condon for *B. subtilis* strains; and S. Albers for sharing her northern blot protocol. We also thank M. Meyer-Jens for the creation of the pHT01 p43 wRBS MCS-GS5-C-Flag tGyrA vector during his laboratory rotation in the D.N.W. laboratory. This research was supported by grants from DFG (WI3285/11-1 to D.N.W.), a Medical Research Council Non-Clinical Senior Research Fellowship (MR/R008205/1 to S.G.) and the Estonian Research Council (PRG335 to T.T.).

**Author contributions** L.D.-P. prepared cryo-EM samples and performed biochemical assays. S.K. and T.T. performed mass spectrometry. S.G. analysed sequencing data and prepared the corresponding figure. H.P. prepared cryo-EM grids and B.B. collected cryo-EM data. L.D.-P. and H.P. processed the microscopy data. H.P. generated and refined the molecular models. L.D.-P., D.N.W. and H.P. analysed and interpreted data. L.D.-P. and H.P. prepared the figures. L.D.-P., D.N.W. and H.P. wrote the manuscript. H.P. conceived and supervised the project.

**Funding** Open access funding provided by Universität Hamburg.

**Competing interests** The authors declare no competing interests.

**Additional information**
**Correspondence and requests for materials** should be addressed to Daniel N. Wilson or Helge Paternoga.

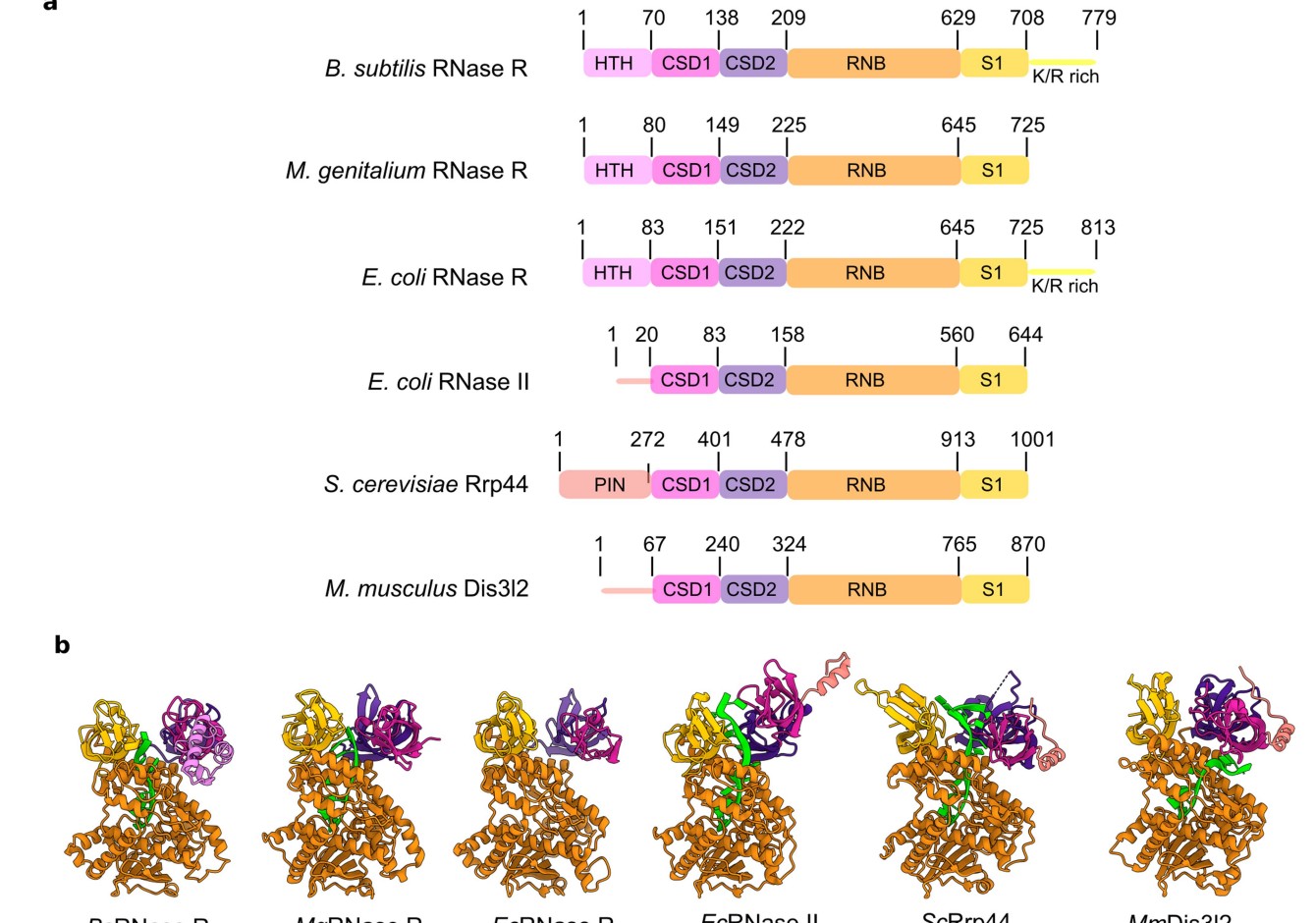

**Extended Data Fig. 1 | RNase R domain arrangement and architecture.**
**a**,**b**, Schematic representation of the domain arrangement (**a**) and molecular structures (**b**) of RNase R from *B. subtilis* (BsRNase R, this study), *M. genitalium* (MgRNase R, PDB ID 7DIC)[18] and *E. coli* (EcRNase R, PDB ID 5XGU)[19], together with RNase II homologues from *E. coli* (EcRNase II, PDB ID 2IX1)[30], *Saccharomyces cerevisiae* Rrp44 (ScRrp44, PDB ID 2VNU)[16] and *Mus musculus* Dis3l2 (MmDis3I2,

PDB ID 4PMW)[31]. The X-ray models of the RNase R homologues are colour-coded as in **a**, and where relevant the ssRNA substrate is shown in green. RNase R domain: HTH; helix-turn-helix domain, CSD; cold-shock domain, RNB; RNase II family 3′ exonuclease domain, S1; S1 ribosomal protein domain, K/R rich; Lys/Arg rich tail, PIN; PIN nuclease domain.

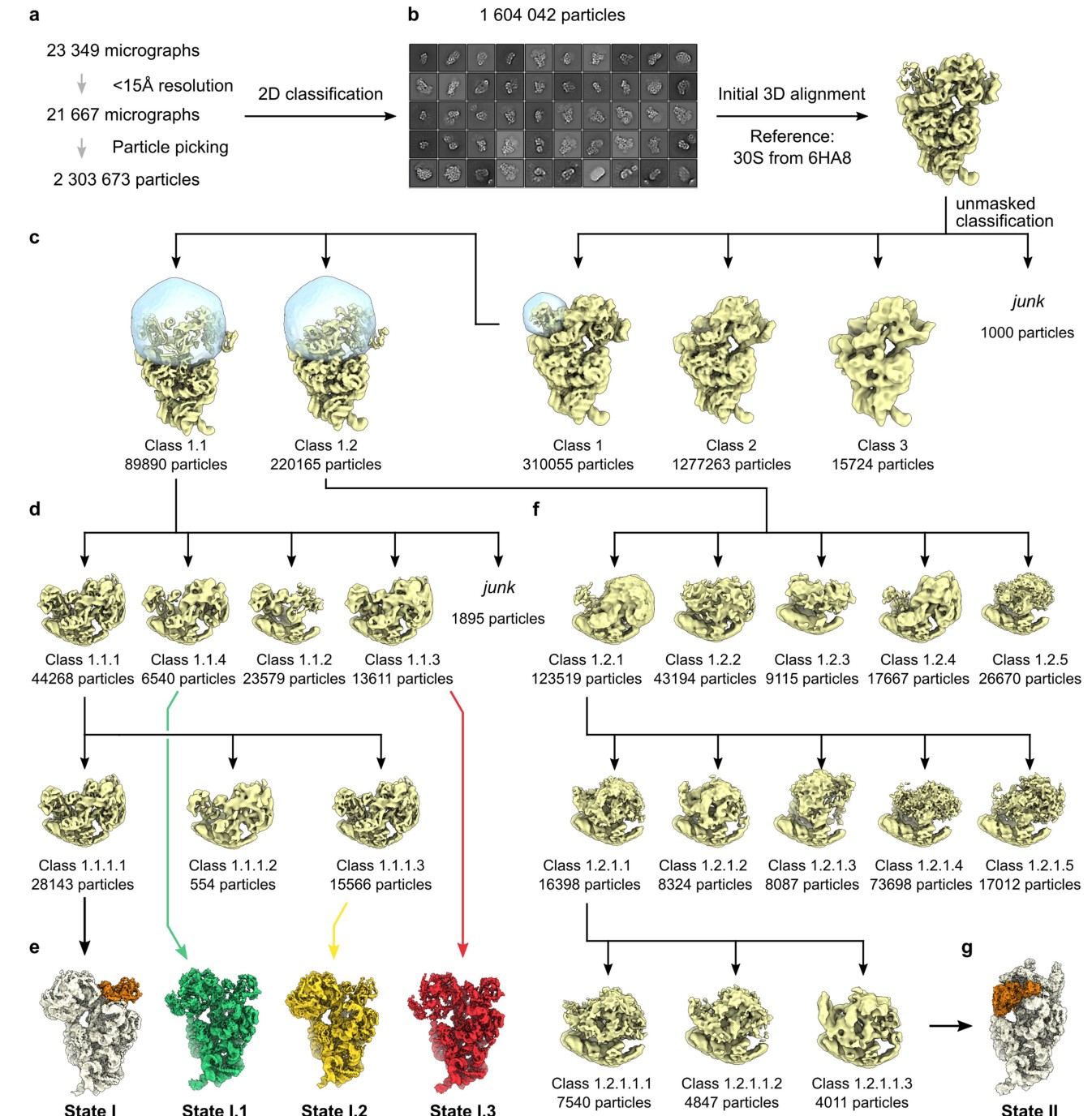

**Extended Data Fig. 2 | Sorting scheme for the RNase R–30S complexes.**
**a**, A total of 2,303,673 particles were picked from 21,667 micrographs and 2D classified. **b**, A total of 1,604,042 particles resembling 30S subunits were selected and 3D refined using the model of a published *B. subtilis* 30S as a reference (PDB ID 6HA8)[33]. **c**, Unmasked 3D classification yielded 3 classes, which were further subsorted: Sorting Class 1 with a mask around the density for RNase R, yielded 2 classes, subclass 1.1 (5.6%) had a clear RNase R density at the mRNA exit channel, whereas Class 2.1 (13.7%) had a less defined density. **d,e**, Two further rounds of subclassification for class 1.1 with a mask encompassing RNase R and a larger portion of the 30S head (**d**) yielded a total of four subclasses (**e**), referred to as state I (1.8%), state I.1 (0.4%), state I.2 (1%) and state I.3 (0.8%). The four maps were post-processed, CTF-refined and polished, yielding final maps with overall resolutions of 3.1 Å, 4.2 Å, 3.6 Å and 3.7 Å, respectively. **f,g**, The second class 1.2 from **c** was subjected to three rounds of subclassification (**f**), which yielded a class, named state II (**g**; 0.25%), which was post-processed, CTF-refined and polished to a final overall resolution of 4.7 Å.

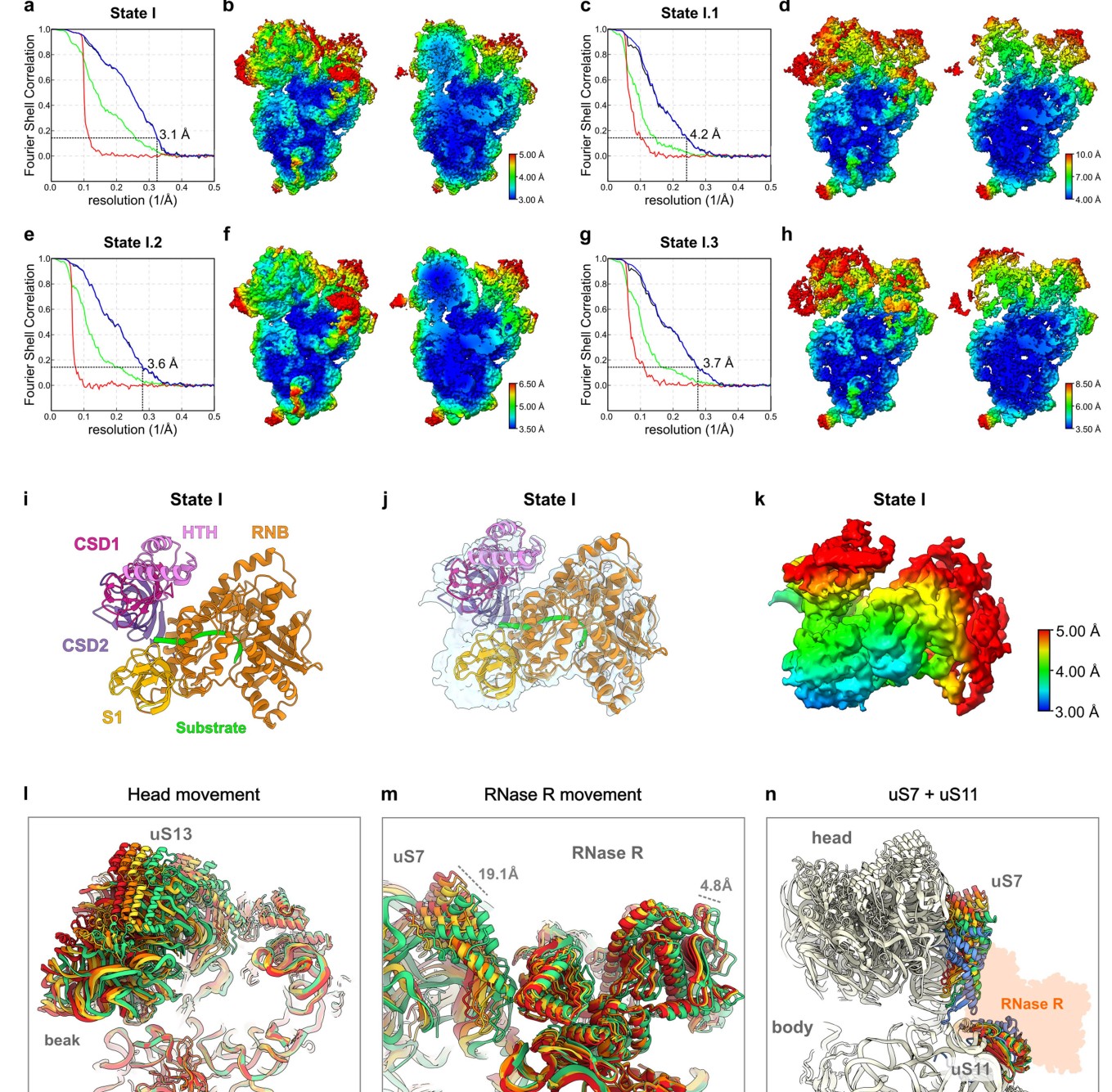

**Extended Data Fig. 3 | State I local resolution and head movement.**
**a**–**h**, Fourier shell correlation (FSC) curves and local resolution of the post-processed RNase R-bound 30S subunits in state I, I.1, I.2 and I.3, respectively. For the FSC curves (**a**,**c**,**e**,**g**), the dashed line at 0.143 indicates an average resolution of 3.1 Å, 4.2 Å, 3.6 Å and 3.7 Å, respectively. The different curves include the masked map (green), unmasked map (blue), the phase-randomized masked map (red) and correlation-corrected curve (black). For the local resolution, overviews (left) and transverse sections (right) of the cryo-EM maps of state I, I.1, I.2 and I.3, respectively are shown (**b**,**d**,**f**,**h**). **i**–**k**, RNase R from state I with model coloured according to its domain boundaries (**i**), model overlaid with cryo-EM map density (**j**) and cryo-EM map density of RNase R coloured according to local resolution (**k**). **l**,**m**, Overlay of state I (orange), state I.1 (green), state I.2 (yellow) and state I.3 (red) with focus on the 30S head (**l**), or on RNase R (**m**). Alignment was based on the 30S body. **n**, Overlay of state I, I.1, I.2, I.3 and the canonical 30S state (PDB ID 6HA8)[33] with uS7 (head) and uS11 (body) coloured in orange, green, yellow, red and blue, respectively.

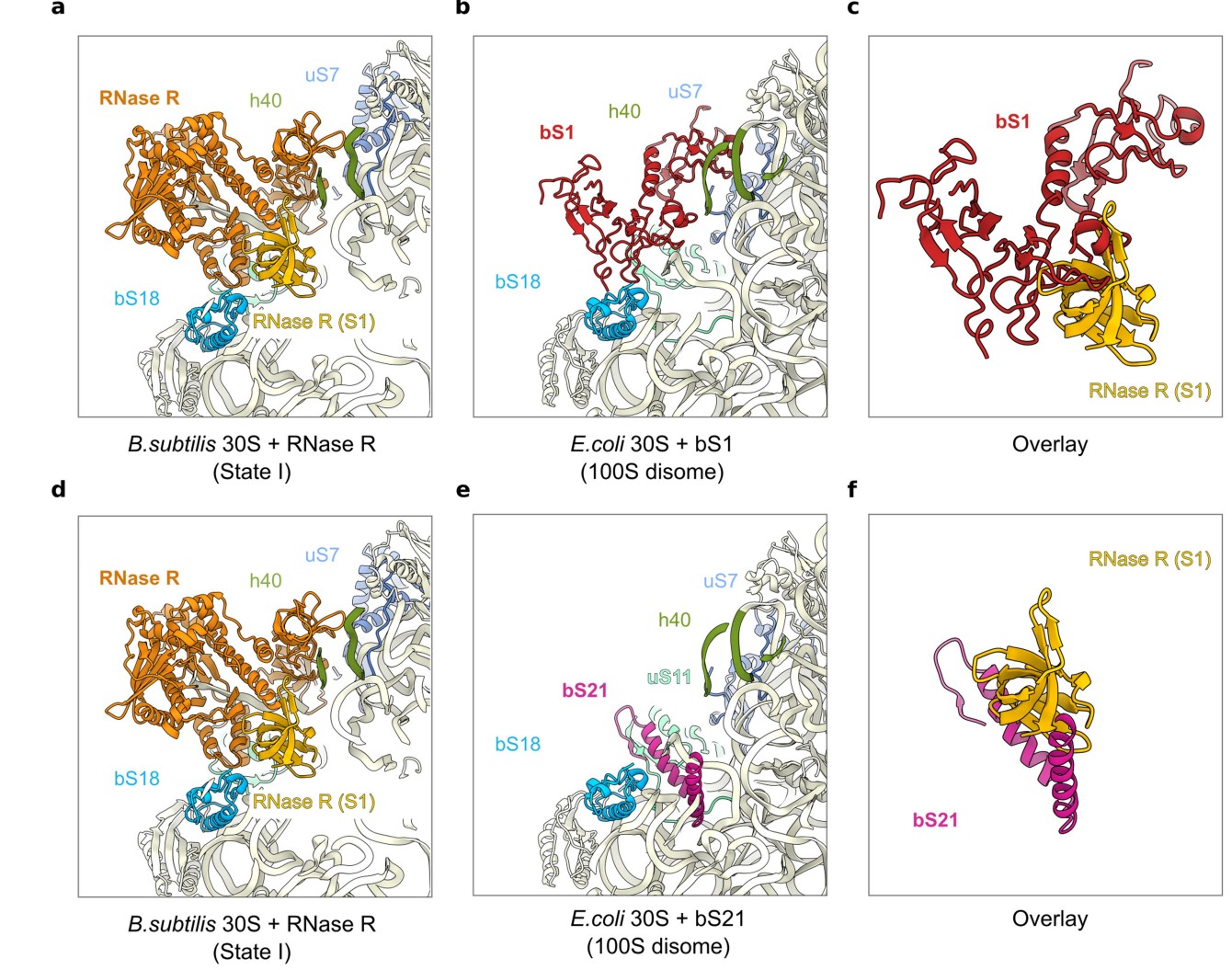

**Extended Data Fig. 4 | Comparison of RNase R with the ribosomal proteins S1 and S21. a**, Model of RNase R (orange) with its S1 domain (gold) as bound to the 30S subunit in state I. **b,c**, bS1 protein (red) bound to the 30S subunit of *E. coli* 100S disome (PDB ID 6H58)[32] (**b**) and overlay of RNase R–S1 domain (gold) and bS1 (red) (**c**) from **a,b. d**, Model of RNase R (orange) with its S1 domain (gold) as bound to the 30S subunit in state I. **e,f**, bS21 protein (magenta) bound to the 30S subunit of *E. coli* 100S disome (PDB ID 6H58)[32] (**e**) and overlay of RNase R–S1 domain (gold) and bS21 (magenta) (**f**) from **a,b**.

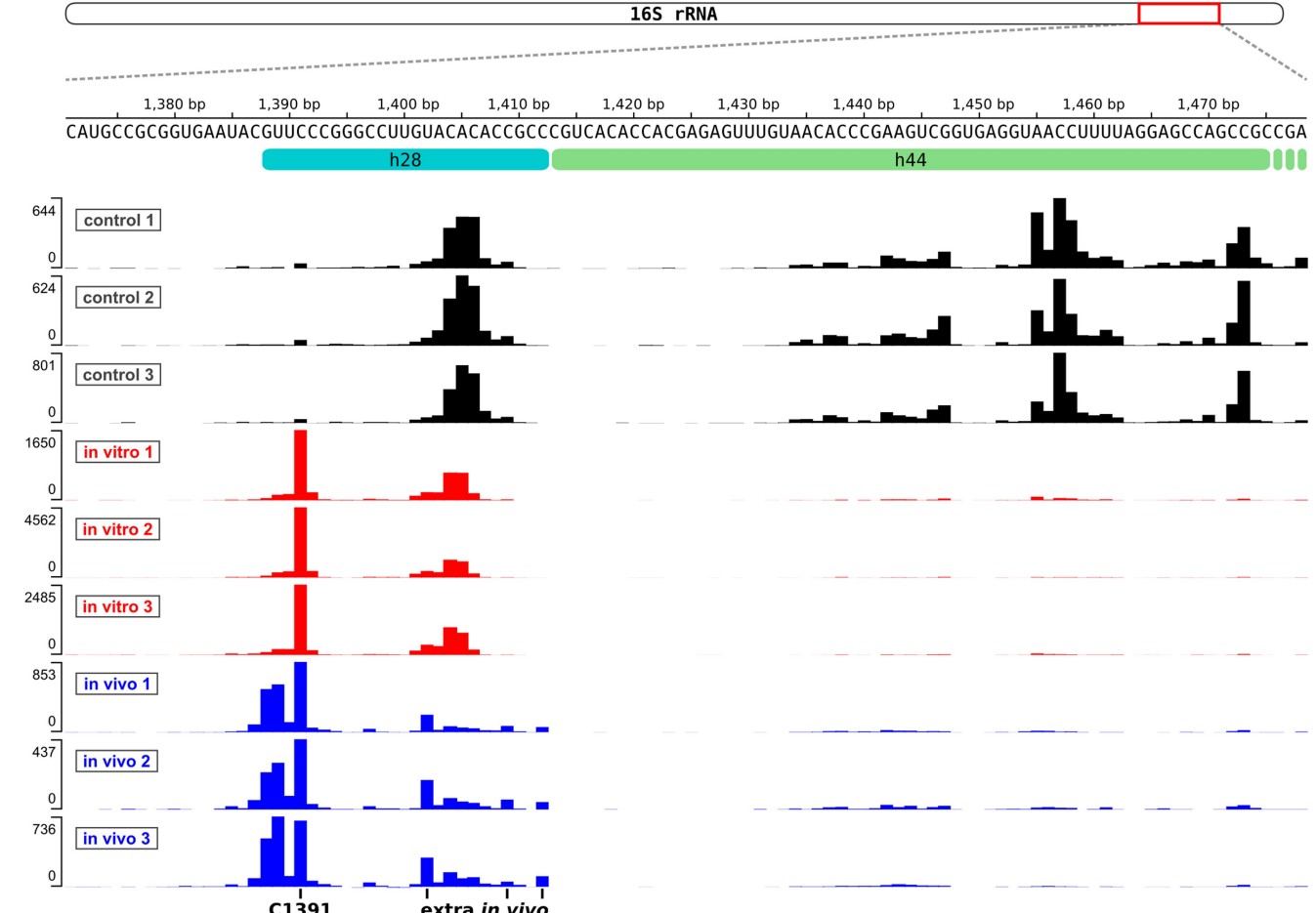

**Extended Data Fig. 5 | Comparison of 16S rRNA 3′ truncation sites between RNase R-mediated in vitro degradation and native pull-outs.** Deep sequencing analysis of RNA species, isolated after in vitro degradation or from native pull-outs. As a control, an in vitro reaction was used that was not incubated at 37 °C. Sequencing hits of 3′ ends of isolated RNAs were plotted against the mature 16S rRNA sequence of *B. subtilis*.

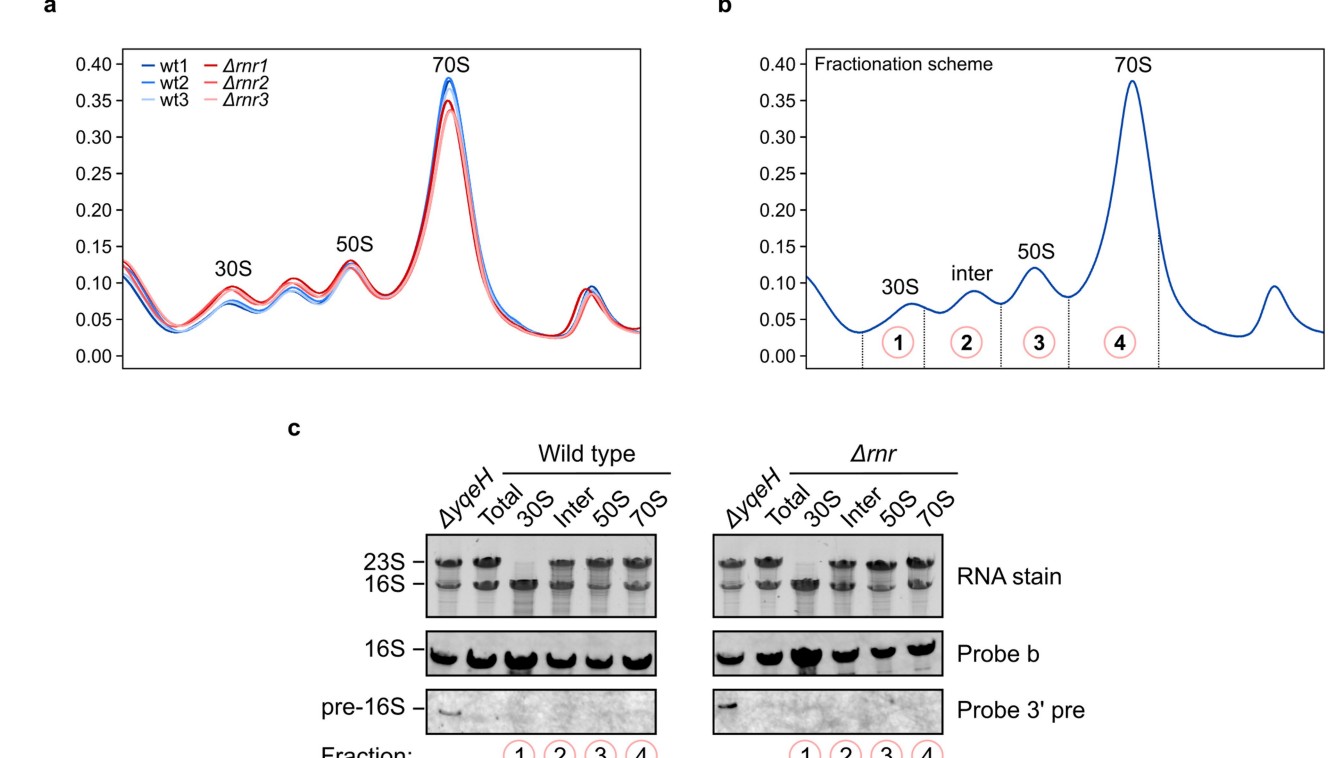

**Extended Data Fig. 6 | Polysome analysis of wild-type and *Δrnr* strains grown to late-exponential phase. a**, Polysome profiles (in triplicate) of sucrose density gradient (10–40% w/v) from wild-type (blue nuances) or *Δrnr* cells (red nuances) (*n* = 3). **b**, A schematic showing the fraction numbering used in **c**. **c**, Northern blot analysis of the RNA extracted from gradient fractions with probes against the mature and pre-16S rRNA. The input was stained with Serva stain G. Total RNA from the input lysates and the *ΔyqeH* strain was used as control. 1 ug of RNA was loaded in all lanes (*n* = 2). For gel source data, see Supplementary Fig. 1.

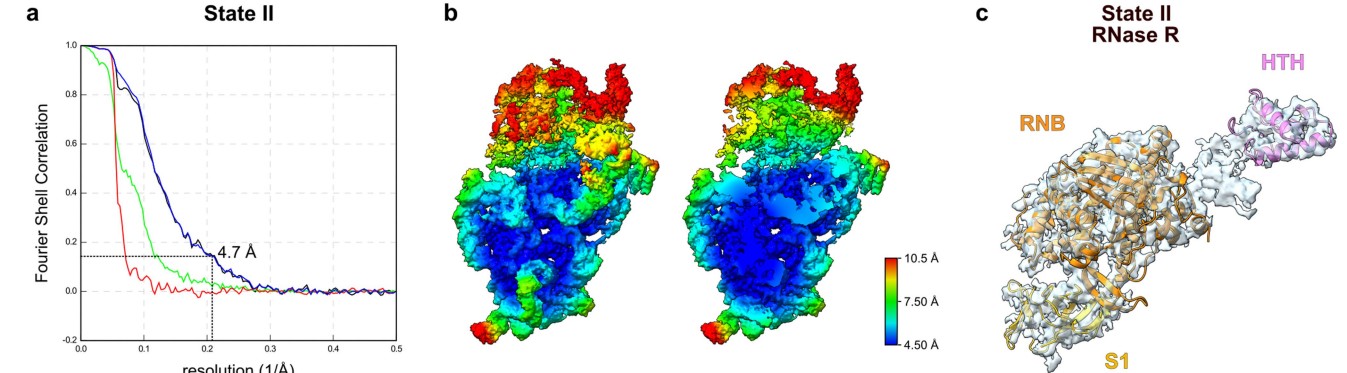

**Extended Data Fig. 7 | Resolution of the state II RNase R–30S complex.**
**a**, FSC curves of the post-processed RNase R–30S complex in state II (masked map in green, unmasked map in blue, phase-randomized masked map in red and correlation-corrected curve in black). The dashed line at $FSC_{0.143}$ indicates an average resolution of 4.7 Å. **b**, Overview (left) and transverse section (right) of the cryo-EM map of state II, coloured according to local resolution. **c**, Isolated cryo-EM density for RNase R within state II with fitted model for RNase R, coloured by domain (HTH, pink; RNB, orange and S1, yellow).

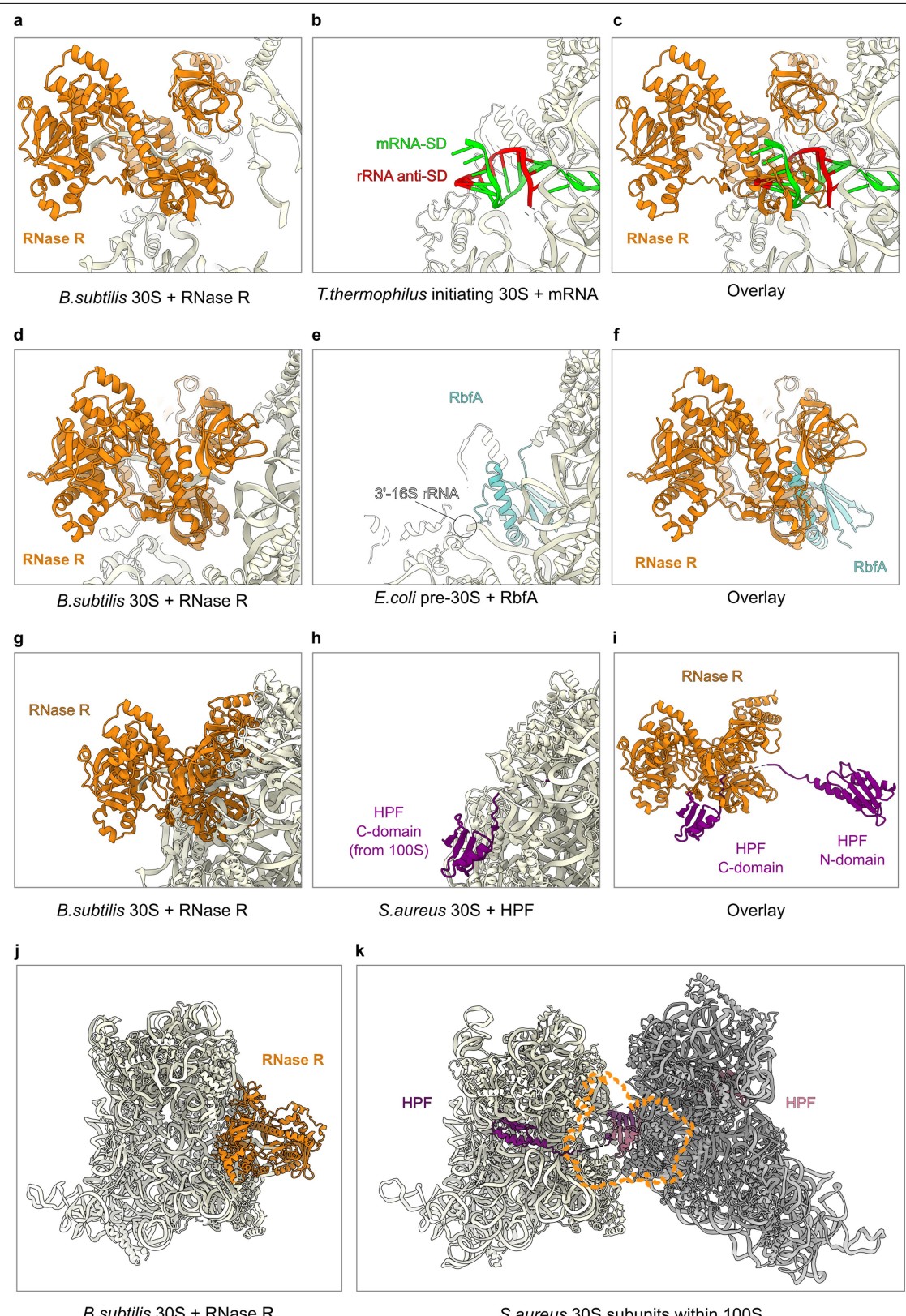

**a** *B.subtilis* 30S + RNase R

**b** *T.thermophilus* initiating 30S + mRNA

**c** Overlay

**d** *B.subtilis* 30S + RNase R

**e** *E.coli* pre-30S + RbfA

**f** Overlay

**g** *B.subtilis* 30S + RNase R

**h** *S.aureus* 30S + HPF

**i** Overlay

**j** *B.subtilis* 30S + RNase R

**k** *S.aureus* 30S subunits within 100S

**Extended Data Fig. 8 | Comparison of the binding site of RNase R in the state I RNase R–30S complex with factors and ligands. a–c**, The binding site of RNase R (orange) in state I RNase R–30S complex (**a**), Shine–Dalgarno-anti-SD with mRNA (green) and 3′ end of 16S rRNA (red) bound to the *Thermus thermophilus* 70S ribosome (PDB ID 5LMN)[48] (**b**) and overlay of RNase R (orange) and SD–anti-SD helix (green-red) (**c**) from **a**,**b**. **d**–**f**, The binding site of RNase R (orange) in state I RNase R–30S complex (**d**), RbfA (cyan) bound to the *E. coli*

pre-30S complex (PDB ID 7BOH)[50] (**e**) and overlay of RNase R (orange) and RbfA (cyan) (**f**) from **d**,**e**. **g**–**i**, The binding site of RNase R (orange) in state I RNase R–30S complex (**g**), HPF (purple) bound to the *S. aureus* 30S within the 100S disome (PDB ID 6FXC)[53] (**h**) and overlay of RNase R (orange) and HPF (purple) (**i**) from **g**,**h**. **j**,**k**, The binding site of RNase R (orange) in state I RNase R–30S complex (**j**), compared with dimerized HPF (purple) and 30S subunits from the *S. aureus* 100S (PDB ID 6FXC)[53] (**k**).

## Extended Data Table 1 | Cryo-EM data collection, refinement and validation statistics

| | State I<br>(EMDB-16595)<br>(PDB 8CDU) | State I.1<br>(EMDB-16606)<br>(PDB 8CED) | State I.2<br>(EMDB-16605)<br>(PDB 8CEC) | State I.3<br>(EMDB-16607)<br>(PDB 8CEE) | State II<br>(EMDB-16596)<br>(PDB 8CDV) |
|---|---|---|---|---|---|
| **Data collection and processing** | | | | | |
| Magnification | 96,000x | 96,000x | 96,000x | 96,000x | 96,000x |
| Acceleration voltage (kV) | 300 | 300 | 300 | 300 | 300 |
| Electron exposure (e⁻/Å²) | 50 | 50 | 50 | 50 | 50 |
| Defocus range (μm) | −0.4 – −0.9 | −0.4 – −0.9 | −0.4 – −0.9 | −0.4 – −0.9 | −0.4 – −0.9 |
| Pixel size (Å) | 0.83 | 0.83 | 0.83 | 0.83 | 0.83 |
| Symmetry imposed | C1 | C1 | C1 | C1 | C1 |
| Initial particle images (no.) | 1,600,131 | 1,600,131 | 1,600,131 | 1,600,131 | 1,600,131 |
| Final particle images (no.) | 28,143 | 6,540 | 15,566 | 13,611 | 4,011 |
| Map resolution (Å) | 3.10 | 4.15 | 3.57 | 3.70 | 4.73 |
| FSC threshold | 0.143 | 0.143 | 0.143 | 0.143 | 0.143 |
| **Refinement** | | | | | |
| Initial model used (PDB code) | 6HA8 | 6HA8 | 6HA8 | 6HA8 | 6HA8 |
| Model resolution (masked, Å) | 3.1 | 4.1 | 3.6 | 3.6 | 4.7 |
| FSC threshold | 0.5 | 0.5 | 0.5 | 0.5 | 0.5 |
| CC (mask) | 0.76 | 0.64 | 0.71 | 0.65 | 0.66 |
| CC (volume) | 0.76 | 0.62 | 0.70 | 0.63 | 0.65 |
| Map sharpening *B* factor (Å²) | -28.9 | -47.8 | -43.4 | -45.3 | -68.9 |
| Model composition | | | | | |
| Non-hydrogen atoms | 51,964 | 51,964 | 51,964 | 51,964 | 43,593 |
| Protein residues | 2,928 | 2,928 | 2,928 | 2,928 | 2,094 |
| RNA residues | 1,340 | 1,340 | 1,340 | 1,340 | 1,258 |
| *B* factors (Å²) | | | | | |
| Protein | 150.40 | 283.03 | 182.31 | 247.22 | 279.00 |
| RNA | 115.33 | 184.16 | 140.19 | 177.51 | 218.92 |
| R.m.s. deviations | | | | | |
| Bond lengths (Å) | 0.009 | 0.009 | 0.010 | 0.009 | 0.009 |
| Bond angles (°) | 1.567 | 1.599 | 1.599 | 1.561 | 1.434 |
| Validation | | | | | |
| MolProbity score | 2.23 | 2.43 | 2.38 | 2.35 | 2.33 |
| Clashscore | 4.94 | 10.23 | 8.32 | 8.64 | 13.20 |
| Poor rotamers (%) | 4.95 | 3.91 | 4.35 | 3.87 | 3.68 |
| Ramachandran plot | | | | | |
| Favored (%) | 93.17 | 92.89 | 93.17 | 93.24 | 96.05 |
| Allowed (%) | 5.96 | 6.38 | 6.10 | 5.89 | 3.02 |
| Disallowed (%) | 0.87 | 0.73 | 0.73 | 0.87 | 0.93 |
| Ramachandran Z-score | -3.44 | -3.32 | -3.58 | -3.17 | -1.91 |

|---|---|

# Reporting Summary

## Statistics

For all statistical analyses, confirm that the following items are present in the figure legend, table legend, main text, or Methods section.

| n/a | Confirmed | |
|---|---|---|
| ☐ | ☒ | The exact sample size (*n*) for each experimental group/condition, given as a discrete number and unit of measurement |
| ☐ | ☒ | A statement on whether measurements were taken from distinct samples or whether the same sample was measured repeatedly |
| ☐ | ☒ | The statistical test(s) used AND whether they are one- or two-sided *Only common tests should be described solely by name; describe more complex techniques in the Methods section.* |
| ☒ | ☐ | A description of all covariates tested |
| ☒ | ☐ | A description of any assumptions or corrections, such as tests of normality and adjustment for multiple comparisons |
| ☐ | ☒ | A full description of the statistical parameters including central tendency (e.g. means) or other basic estimates (e.g. regression coefficient) AND variation (e.g. standard deviation) or associated estimates of uncertainty (e.g. confidence intervals) |
| ☐ | ☒ | For null hypothesis testing, the test statistic (e.g. *F*, *t*, *r*) with confidence intervals, effect sizes, degrees of freedom and *P* value noted *Give P values as exact values whenever suitable.* |
| ☒ | ☐ | For Bayesian analysis, information on the choice of priors and Markov chain Monte Carlo settings |
| ☒ | ☐ | For hierarchical and complex designs, identification of the appropriate level for tests and full reporting of outcomes |
| ☒ | ☐ | Estimates of effect sizes (e.g. Cohen's *d*, Pearson's *r*), indicating how they were calculated |

*Our web collection on statistics for biologists contains articles on many of the points above.*

## Software and code

Policy information about availability of computer code

| Data collection | CryoEM data were collected using the EPU 3.0 software (FEI, Netherlands) |
|---|---|
| Data analysis | RELION v3 with MotionCor2 v1.2.1, CTFFIND 4.1.14, and crYOLO v1.8.0b47 were used for processing micrographs, picking particles, classification and refining cryo-EM maps. RELION v3 was used to calculate local resolution. Coot v0.9.8.5 for model building and ServalCat v0.3.1 with REFMAC 5 v5.8.0403 for model refinement and statistics. Figures were generated using ChimeraX v1.3. |

For manuscripts utilizing custom algorithms or software that are central to the research but not yet described in published literature, software must be made available to editors and reviewers. We strongly encourage code deposition in a community repository (e.g. GitHub). See the Nature Portfolio guidelines for submitting code & software for further information.

## Data

Policy information about availability of data

All manuscripts must include a data availability statement. This statement should provide the following information, where applicable:

- Accession codes, unique identifiers, or web links for publicly available datasets
- A description of any restrictions on data availability
- For clinical datasets or third party data, please ensure that the statement adheres to our policy

Cryo-electron microscopy maps have been deposited at the EMDataBank as follows: StateI: EMD-16595; State I.1: EMD-16606; State I.2: EMD-16605; State I.3: EMD-16607; StateII: EMD-16596. Associated molecular models have been deposited at the ProteinDataBank: State I: PDB-8CDU; State I.1: PDB-8CED; State I.2: PDB-8CEC; StateI.3: PDB-8CEE; State II: PDB-8CDV. The sequencing data can be accessed at NCBI with the code GEO-GSE251701. Published structural data used in this study: ProteinDataBank: PDB-6HA8, PDB-7DIC, PDB-5XGU, PDB-2IX1, PDB-2VNU, PDB-4PMW, PDB-6H58, PDB-5LMN, PDB-7BOH, PDB-6FXC

# Field-specific reporting

Please select the one below that is the best fit for your research. If you are not sure, read the appropriate sections before making your selection.

☒ Life sciences          ☐ Behavioural & social sciences          ☐ Ecological, evolutionary & environmental sciences

For a reference copy of the document with all sections, see nature.com/documents/nr-reporting-summary-flat.pdf

# Life sciences study design

All studies must disclose on these points even when the disclosure is negative.

| | |
|---|---|
| Sample size | No sample size calculation was performed. The sample size was selected on the basis of a three-day data collection, which was chosen to obtain a sufficient number of particles for processing. |
| Data exclusions | Micrographs with low estimated resolution or poorly fitted CTFs were discarded, as were particles that clustered into poorly defined classes during 2D and 3D classification. |
| Replication | All biochemical results shown have been replicated at least twice. n refers to biological replicates, technical replicates are denoted separately. |
| Randomization | Data collection was carried out at regions of the cryo-EM grids that displayed good ice quality and particle density omitting poorer regions. For 3D refinement in RELION, particles are randomly placed in one of two subsets which is done automatically by the software. These subsets are maintained for CTF refinement steps. |
| Blinding | No blinding was performed as blinding is not possible or not applicable for the experiments because the identity of the analyzed sample was known. For biochemical experiments blinding was not relevant as sufficient controls were used. |

# Behavioural & social sciences study design

All studies must disclose on these points even when the disclosure is negative.

| | |
|---|---|
| Study description | Briefly describe the study type including whether data are quantitative, qualitative, or mixed-methods (e.g. qualitative cross-sectional, quantitative experimental, mixed-methods case study). |
| Research sample | State the research sample (e.g. Harvard university undergraduates, villagers in rural India) and provide relevant demographic information (e.g. age, sex) and indicate whether the sample is representative. Provide a rationale for the study sample chosen. For studies involving existing datasets, please describe the dataset and source. |
| Sampling strategy | Describe the sampling procedure (e.g. random, snowball, stratified, convenience). Describe the statistical methods that were used to predetermine sample size OR if no sample-size calculation was performed, describe how sample sizes were chosen and provide a rationale for why these sample sizes are sufficient. For qualitative data, please indicate whether data saturation was considered, and what criteria were used to decide that no further sampling was needed. |
| Data collection | Provide details about the data collection procedure, including the instruments or devices used to record the data (e.g. pen and paper, computer, eye tracker, video or audio equipment) whether anyone was present besides the participant(s) and the researcher, and whether the researcher was blind to experimental condition and/or the study hypothesis during data collection. |
| Timing | Indicate the start and stop dates of data collection. If there is a gap between collection periods, state the dates for each sample cohort. |
| Data exclusions | If no data were excluded from the analyses, state so OR if data were excluded, provide the exact number of exclusions and the rationale behind them, indicating whether exclusion criteria were pre-established. |
| Non-participation | State how many participants dropped out/declined participation and the reason(s) given OR provide response rate OR state that no participants dropped out/declined participation. |
| Randomization | If participants were not allocated into experimental groups, state so OR describe how participants were allocated to groups, and if allocation was not random, describe how covariates were controlled. |

# Ecological, evolutionary & environmental sciences study design

All studies must disclose on these points even when the disclosure is negative.

| | |
|---|---|
| Study description | Briefly describe the study. For quantitative data include treatment factors and interactions, design structure (e.g. factorial, nested, hierarchical), nature and number of experimental units and replicates. |
| Research sample | Describe the research sample (e.g. a group of tagged Passer domesticus, all Stenocereus thurberi within Organ Pipe Cactus National |

| Research sample | *Monument), and provide a rationale for the sample choice. When relevant, describe the organism taxa, source, sex, age range and any manipulations. State what population the sample is meant to represent when applicable. For studies involving existing datasets, describe the data and its source.* |
|---|---|
| Sampling strategy | *Note the sampling procedure. Describe the statistical methods that were used to predetermine sample size OR if no sample-size calculation was performed, describe how sample sizes were chosen and provide a rationale for why these sample sizes are sufficient.* |
| Data collection | *Describe the data collection procedure, including who recorded the data and how.* |
| Timing and spatial scale | *Indicate the start and stop dates of data collection, noting the frequency and periodicity of sampling and providing a rationale for these choices. If there is a gap between collection periods, state the dates for each sample cohort. Specify the spatial scale from which the data are taken* |
| Data exclusions | *If no data were excluded from the analyses, state so OR if data were excluded, describe the exclusions and the rationale behind them, indicating whether exclusion criteria were pre-established.* |
| Reproducibility | *Describe the measures taken to verify the reproducibility of experimental findings. For each experiment, note whether any attempts to repeat the experiment failed OR state that all attempts to repeat the experiment were successful.* |
| Randomization | *Describe how samples/organisms/participants were allocated into groups. If allocation was not random, describe how covariates were controlled. If this is not relevant to your study, explain why.* |
| Blinding | *Describe the extent of blinding used during data acquisition and analysis. If blinding was not possible, describe why OR explain why blinding was not relevant to your study.* |

Did the study involve field work? ☐ Yes ☐ No

## Field work, collection and transport

| Field conditions | *Describe the study conditions for field work, providing relevant parameters (e.g. temperature, rainfall).* |
|---|---|
| Location | *State the location of the sampling or experiment, providing relevant parameters (e.g. latitude and longitude, elevation, water depth).* |
| Access & import/export | *Describe the efforts you have made to access habitats and to collect and import/export your samples in a responsible manner and in compliance with local, national and international laws, noting any permits that were obtained (give the name of the issuing authority, the date of issue, and any identifying information).* |
| Disturbance | *Describe any disturbance caused by the study and how it was minimized.* |

# Reporting for specific materials, systems and methods

We require information from authors about some types of materials, experimental systems and methods used in many studies. Here, indicate whether each material, system or method listed is relevant to your study. If you are not sure if a list item applies to your research, read the appropriate section before selecting a response.

## Materials & experimental systems

| n/a | Involved in the study |
|---|---|
| ☐ | ☒ Antibodies |
| ☒ | ☐ Eukaryotic cell lines |
| ☒ | ☐ Palaeontology and archaeology |
| ☒ | ☐ Animals and other organisms |
| ☒ | ☐ Human research participants |
| ☒ | ☐ Clinical data |
| ☒ | ☐ Dual use research of concern |

## Methods

| n/a | Involved in the study |
|---|---|
| ☒ | ☐ ChIP-seq |
| ☒ | ☐ Flow cytometry |
| ☒ | ☐ MRI-based neuroimaging |

## Antibodies

| Antibodies used | monoclonal Anti-FLAG M2 – HRP antibody; Anti-FLAG M2 Affinity Agarose Gel |
|---|---|
| Validation | monoclonal Anti-FLAG M2-HRP antibody is suitable for the specific detection of FLAG fusion proteins by immunoblotting and is sold for this purpose by the manufacturer. Anti-FLAG M2 affinity gel has been used for the purification of FLAG-fusion proteins and is sold for this purpose by the manufacturer. |

# Eukaryotic cell lines

Policy information about cell lines

| | |
|---|---|
| Cell line source(s) | N/A |
| Authentication | *Describe the authentication procedures for each cell line used OR declare that none of the cell lines used were authenticated.* |
| Mycoplasma contamination | *Confirm that all cell lines tested negative for mycoplasma contamination OR describe the results of the testing for mycoplasma contamination OR declare that the cell lines were not tested for mycoplasma contamination.* |
| Commonly misidentified lines (See ICLAC register) | *Name any commonly misidentified cell lines used in the study and provide a rationale for their use.* |

# Palaeontology and Archaeology

| | |
|---|---|
| Specimen provenance | N/A |
| Specimen deposition | *Indicate where the specimens have been deposited to permit free access by other researchers.* |
| Dating methods | *If new dates are provided, describe how they were obtained (e.g. collection, storage, sample pretreatment and measurement), where they were obtained (i.e. lab name), the calibration program and the protocol for quality assurance OR state that no new dates are provided.* |

☐ Tick this box to confirm that the raw and calibrated dates are available in the paper or in Supplementary Information.

| | |
|---|---|
| Ethics oversight | *Identify the organization(s) that approved or provided guidance on the study protocol, OR state that no ethical approval or guidance was required and explain why not.* |

Note that full information on the approval of the study protocol must also be provided in the manuscript.

# Animals and other organisms

Policy information about studies involving animals; ARRIVE guidelines recommended for reporting animal research

| | |
|---|---|
| Laboratory animals | N/A |
| Wild animals | *Provide details on animals observed in or captured in the field; report species, sex and age where possible. Describe how animals were caught and transported and what happened to captive animals after the study (if killed, explain why and describe method; if released, say where and when) OR state that the study did not involve wild animals.* |
| Field-collected samples | *For laboratory work with field-collected samples, describe all relevant parameters such as housing, maintenance, temperature, photoperiod and end-of-experiment protocol OR state that the study did not involve samples collected from the field.* |
| Ethics oversight | *Identify the organization(s) that approved or provided guidance on the study protocol, OR state that no ethical approval or guidance was required and explain why not.* |

Note that full information on the approval of the study protocol must also be provided in the manuscript.

# Human research participants

Policy information about studies involving human research participants

| | |
|---|---|
| Population characteristics | N/A |
| Recruitment | *Describe how participants were recruited. Outline any potential self-selection bias or other biases that may be present and how these are likely to impact results.* |
| Ethics oversight | *Identify the organization(s) that approved the study protocol.* |

Note that full information on the approval of the study protocol must also be provided in the manuscript.

# Clinical data

Policy information about clinical studies

All manuscripts should comply with the ICMJE guidelines for publication of clinical research and a completed CONSORT checklist must be included with all submissions.

| | |
|---|---|
| Clinical trial registration | N/A |
| Study protocol | *Note where the full trial protocol can be accessed OR if not available, explain why.* |

| Data collection | *Describe the settings and locales of data collection, noting the time periods of recruitment and data collection.* |
|---|---|
| Outcomes | *Describe how you pre-defined primary and secondary outcome measures and how you assessed these measures.* |

# Dual use research of concern

Policy information about dual use research of concern

## Hazards

Could the accidental, deliberate or reckless misuse of agents or technologies generated in the work, or the application of information presented in the manuscript, pose a threat to:

No | Yes
⊠ ☐ Public health
⊠ ☐ National security
⊠ ☐ Crops and/or livestock
⊠ ☐ Ecosystems
⊠ ☐ Any other significant area

## Experiments of concern

Does the work involve any of these experiments of concern:

No | Yes
⊠ ☐ Demonstrate how to render a vaccine ineffective
⊠ ☐ Confer resistance to therapeutically useful antibiotics or antiviral agents
⊠ ☐ Enhance the virulence of a pathogen or render a nonpathogen virulent
⊠ ☐ Increase transmissibility of a pathogen
⊠ ☐ Alter the host range of a pathogen
⊠ ☐ Enable evasion of diagnostic/detection modalities
⊠ ☐ Enable the weaponization of a biological agent or toxin
⊠ ☐ Any other potentially harmful combination of experiments and agents

# ChIP-seq

## Data deposition

☐ Confirm that both raw and final processed data have been deposited in a public database such as GEO.

☐ Confirm that you have deposited or provided access to graph files (e.g. BED files) for the called peaks.

| Data access links
*May remain private before publication.* | N/A |
|---|---|
| Files in database submission | *Provide a list of all files available in the database submission.* |
| Genome browser session
(e.g. UCSC) | *Provide a link to an anonymized genome browser session for "Initial submission" and "Revised version" documents only, to enable peer review. Write "no longer applicable" for "Final submission" documents.* |

## Methodology

| Replicates | *Describe the experimental replicates, specifying number, type and replicate agreement.* |
|---|---|
| Sequencing depth | *Describe the sequencing depth for each experiment, providing the total number of reads, uniquely mapped reads, length of reads and whether they were paired- or single-end.* |
| Antibodies | *Describe the antibodies used for the ChIP-seq experiments; as applicable, provide supplier name, catalog number, clone name, and lot number.* |
| Peak calling parameters | *Specify the command line program and parameters used for read mapping and peak calling, including the ChIP, control and index files used.* |
| Data quality | *Describe the methods used to ensure data quality in full detail, including how many peaks are at FDR 5% and above 5-fold enrichment.* |

| Software | *Describe the software used to collect and analyze the ChIP-seq data. For custom code that has been deposited into a community repository, provide accession details.* |
|---|---|

# Flow Cytometry

## Plots

Confirm that:

☐ The axis labels state the marker and fluorochrome used (e.g. CD4-FITC).

☐ The axis scales are clearly visible. Include numbers along axes only for bottom left plot of group (a 'group' is an analysis of identical markers).

☐ All plots are contour plots with outliers or pseudocolor plots.

☐ A numerical value for number of cells or percentage (with statistics) is provided.

## Methodology

| Sample preparation | N/A |
|---|---|
| Instrument | *Identify the instrument used for data collection, specifying make and model number.* |
| Software | *Describe the software used to collect and analyze the flow cytometry data. For custom code that has been deposited into a community repository, provide accession details.* |
| Cell population abundance | *Describe the abundance of the relevant cell populations within post-sort fractions, providing details on the purity of the samples and how it was determined.* |
| Gating strategy | *Describe the gating strategy used for all relevant experiments, specifying the preliminary FSC/SSC gates of the starting cell population, indicating where boundaries between "positive" and "negative" staining cell populations are defined.* |

☐ Tick this box to confirm that a figure exemplifying the gating strategy is provided in the Supplementary Information.

# Magnetic resonance imaging

## Experimental design

| Design type | N/A |
|---|---|
| Design specifications | *Specify the number of blocks, trials or experimental units per session and/or subject, and specify the length of each trial or block (if trials are blocked) and interval between trials.* |
| Behavioral performance measures | *State number and/or type of variables recorded (e.g. correct button press, response time) and what statistics were used to establish that the subjects were performing the task as expected (e.g. mean, range, and/or standard deviation across subjects).* |

## Acquisition

| Imaging type(s) | *Specify: functional, structural, diffusion, perfusion.* |
|---|---|
| Field strength | *Specify in Tesla* |
| Sequence & imaging parameters | *Specify the pulse sequence type (gradient echo, spin echo, etc.), imaging type (EPI, spiral, etc.), field of view, matrix size, slice thickness, orientation and TE/TR/flip angle.* |
| Area of acquisition | *State whether a whole brain scan was used OR define the area of acquisition, describing how the region was determined.* |
| Diffusion MRI | ☐ Used ☐ Not used |

## Preprocessing

| Preprocessing software | *Provide detail on software version and revision number and on specific parameters (model/functions, brain extraction, segmentation, smoothing kernel size, etc.).* |
|---|---|
| Normalization | *If data were normalized/standardized, describe the approach(es): specify linear or non-linear and define image types used for transformation OR indicate that data were not normalized and explain rationale for lack of normalization.* |
| Normalization template | *Describe the template used for normalization/transformation, specifying subject space or group standardized space (e.g. original Talairach, MNI305, ICBM152) OR indicate that the data were not normalized.* |

| | |
|---|---|
| Noise and artifact removal | *Describe your procedure(s) for artifact and structured noise removal, specifying motion parameters, tissue signals and physiological signals (heart rate, respiration).* |
| Volume censoring | *Define your software and/or method and criteria for volume censoring, and state the extent of such censoring.* |

## Statistical modeling & inference

| | |
|---|---|
| Model type and settings | *Specify type (mass univariate, multivariate, RSA, predictive, etc.) and describe essential details of the model at the first and second levels (e.g. fixed, random or mixed effects; drift or auto-correlation).* |
| Effect(s) tested | *Define precise effect in terms of the task or stimulus conditions instead of psychological concepts and indicate whether ANOVA or factorial designs were used.* |

Specify type of analysis: ☐ Whole brain ☐ ROI-based ☐ Both

| | |
|---|---|
| Statistic type for inference<br>(See Eklund et al. 2016) | *Specify voxel-wise or cluster-wise and report all relevant parameters for cluster-wise methods.* |
| Correction | *Describe the type of correction and how it is obtained for multiple comparisons (e.g. FWE, FDR, permutation or Monte Carlo).* |

## Models & analysis

n/a | Involved in the study
☐ | ☐ Functional and/or effective connectivity
☐ | ☐ Graph analysis
☐ | ☐ Multivariate modeling or predictive analysis

| | |
|---|---|
| Functional and/or effective connectivity | *Report the measures of dependence used and the model details (e.g. Pearson correlation, partial correlation, mutual information).* |
| Graph analysis | *Report the dependent variable and connectivity measure, specifying weighted graph or binarized graph, subject- or group-level, and the global and/or node summaries used (e.g. clustering coefficient, efficiency, etc.).* |
| Multivariate modeling and predictive analysis | *Specify independent variables, features extraction and dimension reduction, model, training and evaluation metrics.* |

