## [Peer Review File · Nature]

Manuscript Title: Structural basis of ribosomal 30S subunit degradation by RNase R

Reviewer Comments & Author Rebuttals

Reviewer Reports on the Initial Version:

Referees' comments:

Referee #1 (Remarks to the Author):

Structural insights into the degradation of ribosomes have been lacking, and in this manuscript the authors report the native structures of two distinct small ribosomal 30S subunit degradation intermediates from *B. subtilis* associated with ribonuclease R. The mechanism proposed indicates that RNase R targets orphaned 30S subunits using a dynamic anchored binding site switching mechanism.

The association of RNase R with 30S subunits and 70S ribosomes has been previously reported; however, it earlier work proposed that endonucleolytic cleavages would precede/help the action of the exoribonuclease RNase R. Here it is demonstrated that RNase R is able to degrade 30S and 30 S degradation intermediates without the need of endonucleases, establishing interactions with the head and body of the 30S (even though the authors do not exclude that in some cases other RNases can also act); RNase R predominantly uses domains located in the N-terminal portion of the molecule and, by itself, is capable of initiating decay and fully degrade the 16S rRNA within the context of a mature 30S subunit. The enzyme shows a preference of 30S over 50S.

Another novelty of this work is that it shows that RNase R binds initially at a site located between the 30S head and platform, where the mRNA exit site is located, and later induces a major conformational re-arrangement in the position of the 30S head. The RNase R parallels this movement, relocating to the decoding site by using its N-terminal helix-turn-helix (HTH), which likely serves as an anchor through its contacts.

The approach taken for the structural information was based on cryo-EM, the quality of the data is very good and well presented.

I think that the work is very interesting and new, and I only ask for minor modifications. In light of their new results the authors should also discuss what has already been demonstrated for RNase R (see below)

Minor Comments:

- In page 3 you say " In *Escherichia coli*, triple deletion of RNase R, RNase II and a third exonuclease, polynucleotide phosphorylase (PNPase), leads to the accumulation of truncated rRNA products" This is not correct. You can not have a triple deletion because it is not viable! Double deletion PNPase RNase II is not possible: In 1986 it was the first time that it was shown that you could not make a double mutant deficient in PNPase and RNase II (Donovan WP, Kushner SR. Polynucleotide phosphorylase and ribonuclease II are required for cell viability and mRNA turnover in *Escherichia coli* K-12. *Proc Natl Acad Sci U S A*. 1986 Jan;83(1):120-4. doi: 10.1073/pnas.83.1.120.) – please refer. And later several others have shown that a deletion PNPase and RNase R was not viable. Please

correct your sentence taking all this into account.

- RNase R has been shown to be a processive enzyme. You mention that it is processive but you do not show/discuss how this aspect is important for its activity. For instance, enzymes of the same family (like RNaseII) can be processive and then become distributive, when the substrate is smaller than 10nt. (Frazão et al, Nature 2006).

- Transcriptome sequencing (RNA-seq) analysis of *B. subtilis* strains lacking RNase R suggested that this enzyme did not play a major role in mRNA turnover in the wild-type strain (Chhabra S, et al, mBio 2022). Can you propose an explanation, taking into account your work?

- It has been shown that RNase R deprived of its RNA binding domains, can degrade blunt double-stranded RNA (even though with little efficiency); showing that the RNA-binding domains can select the RNAs with an overhang, and then target them for degradation (Matos RG et al, Biochem J., 2009). Can you discuss this, based on your model?

- You briefly mention the role of RNase R in quality control. In the paper "RNA quality control: degradation of defective transfer RNA. Li Z, Reimers S, Pandit S, Deutscher MP. EMBO J. 2002, propose that defective stable RNA precursors that are poorly converted to their mature forms may be polyadenylated and subsequently degraded. Furthermore other papers show that poly A can be important for RNase R degradation. Can you please further comment RNase R and polyadenylation?

Referee #2 (Remarks to the Author):

Protein synthesis is an essential process in the cell, and requires delicate regulation for ribosome generation, maturation and degradation. Many structural studies have been focused in this field. However, structural insight into the degradation of ribosomes has been lacking. In this paper, the authors report cryo-EM structures of two distinct small ribosomal 30S subunit associated with the 3' to 5' exonuclease-RNase R in two distinct degradation intermediate states. One structure represents the initial binding of RNase R to 30S to facilitate degradation of the anti-Shine-Dalgarno sequence, and the other reveals a large conformational change both in 30S and RNase R from the initial position. These structures provide a mechanistic basis for RNase R-mediated 30S degradation and suggest that RNase R targets orphaned 30S using a dynamic anchored binding site switching mechanism. Overall, the topic is important and the structural data are solid with good quality. MY major concern is the biochemical data and functional analyses, they do not match the quality of the structural data and do not provide enough supports as they should for the conclusions from the structures.

Comments:

In the paper, the authors mentioned the secondary structures of 30S many times, such as h40, h44 and h45. Given the broad range of the audience, there should at least be a schematic picture of the 30S RNA with all the major secondary structures labeled, so that readers could easily understand

what are the authors' points in the figures.

Fig. 2 e and f. The authors employed in vitro degradation assay and biochemical mapping to find out the truncation site in the first degradation intermediate state. Since this is an in vitro purified system, can the authors isolate the products and use sequencing method to figure out the exact sequence of 30S in this state?

Figure 3. The authors used both in vitro degradation assay (a-e) and in vivo composition analysis to study the importance of RNase R. Given that the authors now have all the detailed structural information of the 30S-RNase R complex, they should design some key point mutations based on the structure to understand the functional important of the essential RNA-protein interfaces revealed by the structure. This would be much stronger than the current data which do not actually require the structural information. This is especially true for the case of Figure 3f, whose current form does not support the important value of the structure. And again, the authors could use sequencing method to find out the exact sequence of the products (Figure 3b 3d).

Figure 1c. It is hard to identify the N and C termini. They have the same color.

Figure 1d, CSD1 is not labeled.

Referee #3 (Remarks to the Author):

Dimitrova-Paternoga et al. describe among the first structural insights into ribosome subunit degradation, specifically *Bacillus subtilis* 30S rRNA interaction with RNase R. They perform cryogenic electron microscopy (cryo-EM) analysis of RNase R-30S complexes immunoprecipitated from *Bacillus subtilis* lysates via RNase R-FLAG overexpression, deriving two primary/global structural states by multiparticle sorting/refinement. These structures visualize direct RNase R interaction with truncated 3' 16S rRNA, in 30S states with unprecedented large-scale domain rearrangements, and lacking ribosomal proteins, both of which are distinct from previously solved mature 30S structures. They substantiate their structural data with biochemical analysis of in vitro mature 30S degradation by recombinant RNase R, and ex vivo 30S accumulation in an RNase R deletion (Δ rnrr) strain. The authors ultimately conclude from these data that RNase R degrades mature 30S made vulnerable by the lack of complexation with mRNA and/or 50S subunits. The authors speculate in the Discussion that RNase R may degrade 30S subunits in conditions where its homeostasis is necessary, such as stationary growth, starvation, and/or quality control when the subunit is damaged or stalled during translation.

The Reviewer finds the study and its results of high interest, making a unique contribution to the ribosome field that has major gaps its understanding of ribosome degradation. The structural and biochemical data are solid, rationalizing a mechanism of 16S rRNA degradation by RNase R. The major issues to be addressed in revision principally concern the authors' interpretation of the data, with regard to the 30S states targeted by RNase R in vivo. In particular, it will be important for the authors to disentangle two possible interpretations of their data: the biological function of RNase R

is to degrade mature 30S, and/or RNase R degrades immature 30S assembly precursors. The following are detailed comments geared towards clarifying these two possibilities, thus linking the excellent structural data to a biologically solid interpretation.

1.0. (major) The authors' interpretation of the structural data is that RNase R degrades "orphaned" (nice term) "mature" 30S, as summarized in the Discussion (lines 314-316; Figure 5). RNase R-30S structures are reported to be absent for ribosomal proteins bS21 (lines 135-136), uS2, and uS3 (lines 287-289). The absence of these ribosomal proteins is interpreted to be a consequence of RNase R-mediated displacement, or their dissociation is a prerequisite for RNase R binding (lines 136-138). However, complicating this interpretation is that precisely bS21, uS2, and uS3 are absent in the latest-stage 30S assembly intermediates (eg. assembly map from Chen & Williamson, JMB 2013, <http://dx.doi.org/10.1016/j.jmb.2012.11.040>, Figure 4a). This is substantiated by emerging work from the Davis and Ortega labs, which includes structural analysis of 30S assembly in the absence of KsgA, a biogenesis methyltransferase (<https://doi.org/10.1101/2022.07.13.499473>). It's notable in this second paper that bS21 and uS2 appear to be particularly affected in Δ ksgA cells, and that the map density for the 30S head is highly fragmented. The Reviewer appreciates that the biochemical data with purified mature 30S supports the idea that reconstitution with RNase R can degrade orphaned mature 30S in vitro – these data are solid. Nonetheless, the above issues raised in the overexpression-based ex vivo-derived structures is whether RNase R does degrade orphaned "mature" 30S in vivo. This issue is detailed further in several related Reviewer comments, as follows.

1.1. (major) Mass spectrometry was used to confirm co-immunoprecipitation of ribosomal particles with RNase R-FLAG (line 86), with the authors noting the enrichment of ribosomal proteins. Notably, in the related Supplementary Data table, many subunit biogenesis factors appear to co-purify with RNase R-FLAG at levels comparable to ribosomal proteins themselves (eg. CshA, YqeI, RnjB, RsfS, RnjA, RimM, RlmCD, TrmB, InfC, EngB, RsmB, DbpA, Obg). Complicating the picture is that many of the ribosomal proteins are 50S (rather than 30S), and some of these biogenesis factors are 50S-associated (eg. RsfS, InfC, DbpA, Obg). These observations are apparent when rank-sorting the table by the average iBAQ of RNase R-FLAG IP replicates, and also when rank-sorting the RNase R-FLAG average normalized to the iBAQ control. As the authors raise the "question as to how mature 30S subunits protect themselves from RNase R action in the actively growing cell" (lines 221-222), it is thus worth considering the action of biogenesis factors in protecting assembly intermediates, in addition to mRNA and the 50S.

1.2. (major) Sucrose density gradients analyzed subunits in WT vs. Δ rnr cells (Fig. S7), with the interpretation that loss of RNase R leads to the accumulation of 30S relative to the 70S (Fig. 3f). The authors are encouraged to overlay the gradient profiles in Fig. S7 to facilitate direct comparison, since the y-axis limits are not consistent. Regardless, it would appear that while there is a legitimate accumulation of 30S (whether compared against 70S or 50S), the 70S itself appears decreased in the Δ rnr strain compared to WT. Given the questions regarding 30S assembly intermediates raised above, it may be that the accumulation of 30S represents 70S-incompetent precursors that would normally be degraded by RNase R. The consistent appearance of a peak with intermediate sedimentation between the 30S and 50S in both WT and Δ rnr begs the question of what these peaks actually represent. The authors should run an RNA gel derived from serially collected gradient fractions to evaluate the identity of the peaks. The authors should describe in the Methods at what

growth stage the cells were collected before lysis in this experiment, as this additional peak may be subunit degradation during the transition from logarithmic to stationary phase (mediated by many RNAses), as described by the Murray Deutscher lab, for example. Finally, statistical significance testing of mean differences between quantified peaks must be presented in Fig. 3f (and legend), with individual replicate values and dispersion represented in the graph.

1.3. (suggestions) The authors are encouraged to address Comments 1.0-1.2 as they see fit with biochemical/phenotyping assays - the Reviewer does not expect further labor-intensive structural experiments. Some combination of the following experiments is suggested to disentangle the orphaned "mature" vs. "assembly precursor" 30S targets of RNase R. (1) Knock-in RNase R-FLAG, followed by IP-mass spectrometry to assess the binding partners in more physiologic conditions. (2) Generate orphaned mature 30S by overexpression of a 70S splitting factor or anti-association factor (eg. ObgE, RsfS), which should accumulate free mature 30S, and see if RNase R is enriched or selectively associates with these abnormally accumulating 30S (eg. by sucrose gradient with RNA gel of the fractions). (3) Antisense-knockdown or knockout of biogenesis factors (eg. KsgA, RbfA) to generate precursor 30S, and likewise analyze if RNase R enriches with these precursors. (4) Culture in stress conditions or induce stalling, similar to the speculated conditions of RNase R-mediated 30S homeostasis described in the Discussion, where RNase R should be recruited to specifically mature vs. precursor 30S. Performing RNase R-IP at earlier growth time points, such as early log phase (OD₆₀₀= 0.3), or mid log phase (OD₆₀₀= 0.5), may also be informative. Of course, it may be that ultimately RNase R targets both mature and precursor 30S, but these data would critically facilitate interpretation of the underlying 30S structures and physiologic role of RNase R.

2.0. (minor) The authors' approach to purify RNase R-ribosome complexes for cryo-EM appears to involve RNase R-FLAG overexpression in *Bacillus subtilis* with an IPTG-inducible ectopic expression vector (pHT01), rather than an alternative approach such as knock-in to the native locus. This might not be immediately clear to a non-specialist reader in the Results or Methods, and should be explicitly described with more detail, warranted in the Methods especially, such as: What is IPTG protocol for expression of pHT01? At what temperature and shaking are the cells grown?

2.1. (minor) The purification approach (Comment 2.0) for structural analysis impacts the interpretation of the structural data. The Reviewer applauds the *ex vivo*-derived approach, and appreciates that overexpression of RNase R is likely to be helpful, maybe even critical, to enrich sufficient material for cryo-EM (rather than physiologic expression from the native locus). However, the potential complication is that super-stoichiometric RNase R may accumulate in a non-physiologic manner on complexes normally protected robustly by biogenesis factors. Indeed, then the physiologic role of RNase R may be to act on orphaned mature 30S, as the authors expect from their biochemistry, but the specific approach could derail this interpretation with structures potentially representative of late-stage assembly precursors. Thus, addressing this limitation in the Discussion, in addition to addressing the major points above, would clarify the extent to which these data reflect RNase R's physiologic role.

2.2. (minor) The Methods description of the mass spectrometry is vague as to what samples were analyzed. It would appear from the table that purified 30S and 70S (presumably purified by sucrose gradients?) were shot for comparison. Clarification in the Methods text is required.

2.3. (minor) The authors are encouraged to cite Failmezger et al. 2016 (<https://doi.org/10.1371/journal.pone.0168764>), which supports their interpretation regarding helix 44 and targeted 30S degradation.

Author Rebuttals to Initial Comments:

Referees' comments:

Referee #1 (Remarks to the Author):

Structural insights into the degradation of ribosomes have been lacking, and in this manuscript the authors report the native structures of two distinct small ribosomal 30S subunit degradation intermediates from *B. subtilis* associated with ribonuclease R. The mechanism proposed indicates that RNase R targets orphaned 30S subunits using a dynamic anchored binding site switching mechanism.

The association of RNase R with 30S subunits and 70S ribosomes has been previously reported; however, it earlier work proposed that endonucleolytic cleavages would precede/help the action of the exoribonuclease RNase R. Here it is demonstrated that RNase R is able to degrade 30S and 30 S degradation intermediates without the need of endonucleases, establishing interactions with the head and body of the 30S (even though the authors do not exclude that in some cases other RNases can also act); RNase R predominantly uses domains located in the N-terminal portion of the molecule and, by itself, is capable of initiating decay and fully degrade the 16S rRNA within the context of a mature 30S subunit. The enzyme shows a preference of 30S over 50S.

Another novelty of this work is that it shows that RNase R binds initially at a site located between the 30S head and platform, where the mRNA exit site is located, and later induces a major conformational re-arrangement in the position of the 30S head. The RNase R parallels this movement, relocating to the decoding site by using its N-terminal helix-turn-helix (HTH), which likely serves as an anchor through its contacts.

The approach taken for the structural information was based on cryo-EM, the quality of the data is very good and well presented.

I think that the work is very interesting and new, and I only ask for minor modifications. In light of their new results the authors should also discuss what has already been demonstrated for RNase R (see below)

Minor Comments:

- In page 3 you say " In *Escherichia coli*, triple deletion of RNase R, RNase II and a third exonuclease, polynucleotide phosphorylase (PNPase), leads to the accumulation of truncated rRNA products" This is not correct. You can not have a triple deletion because it is not viable! Double deletion PNPase RNase II is not possible: In 1986 it was the first time that it was shown that you could not make a double mutant deficient in PNPase and RNase II (Donovan WP, Kushner SR. Polynucleotide phosphorylase and ribonuclease II are required for cell viability and mRNA turnover in *Escherichia coli* K-12. *Proc Natl Acad Sci U S A*. 1986 Jan;83(1):120-4. doi: 10.1073/pnas.83.1.120.) – please refer. And later several others have shown that a deletion PNPase and RNase R was not viable. Please correct your sentence taking all this into account.

The reviewer is correct, deletion of RNase R and PNPase or RNase II and PNPase is lethal to the cell. We have now made this clearer in the text on page 3. We have also amended our sentence to make it clear that it is not the triple deletion of RNase R, RNase II and PNPase (as we had incorrectly stated) but rather the deletion of RNase R and RNase II combined with temperature-sensitive mutants of PNPase that leads to accumulation of truncated rRNA products.

- RNase R has been shown to be a processive enzyme. You mention that it is processive but you do not show/discuss how this aspect is important for its activity. For instance, enzymes of the same family (like RNaseII) can be processive and then become distributive, when the substrate is smaller than 10nt. (Frazão et al, Nature 2006).

This is a good point raised by the reviewer that we did not directly address in the initial manuscript. Although our data do not provide direct insight into the importance of the processivity of RNase R for its degradation activity, our data suggest that RNase R must dissociate and rebind to overcome the roadblock between states I and II, suggesting a mix of processive and distributive activity of RNase R is important for complete 30S degradation. This is now mentioned in the discussion on page 12.

- Transcriptome sequencing (RNA-seq) analysis of B. subtilis strains lacking RNase R suggested that this enzyme did not play a major role in mRNA turnover in the wild-type strain (Chhabra S, et al, mBio 2022). Can you propose an explanation, taking into account your work?

Chhabra et al demonstrate that RNase R has a negligible role in mRNA turnover during exponential growth which suggests that the enzyme is either inactive or active on other RNAs during this growth condition. It is well established that RNase R is able to turnover ribosomal subunits in stationary phase and our work shows association of RNase R with 30S subunits purified from late exponential phase cells. In addition, it has been shown in E. coli that RNase R engages mRNAs during cold acclimation (reviewed in Zhang and Gross "Cold Shock Response in Bacteria." Annual Review of Genetics 55, no. 1 (November 23, 2021): 377–400). Therefore, the minor effect on mRNA degradation during exponential growth might indicate a reduced need for the turnover of structured RNAs in this growth phase. However, such statements would require additional experimental support that we feel is tangential to the focus of our study.

- It has been shown that RNase R deprived of its RNA binding domains, can degrade blunt double-stranded RNA (even though with little efficiency); showing that the RNA-binding domains can select the RNAs with an overhang, and then target them for degradation (Matos RG et al, Biochem J., 2009). Can you discuss this, based on your model?

In our model, RNase R recognizes the 30S subunit adjacent to the 3' end of 16S rRNA. The interaction is not directly mediated by the 3' end substrate, but also by neighboring regions of the small subunit. This is different from previous studies where the enzyme digested smaller RNAs that were directly bound and fed towards the active site. From our analysis and previous work, it is clear that RNase R requires a single-stranded overhang to initiate the turnover reaction. This is evident from our in vitro assays of 50S ribosomal subunits showing that the 23S rRNA in the context of 50S subunit is fully protected from RNase R action, which can be explained by the fact that the 23S rRNA does not present a single-stranded 3' extension, being base-paired with the 5' end. Our analysis also shows that the enzyme is unable to move the RNA-binding domains away to allow access of the RNB domain directly to the 3' end for engagement of blunt double-stranded substrates. In addition, we have performed experiments with RNase R mutants that are directed at RNA interaction regions identified by our RNase R-30S complex (Fig. 3f). This analysis shows that the HTH domain in addition to the RNB domain is required for optimal association of RNase R with 30S subunits during turnover.

- You briefly mention the role of RNase R in quality control. In the paper "RNA quality control: degradation of defective transfer RNA. Li Z, Reimers S, Pandit S, Deutscher MP. EMBO J. 2002, propose that defective stable RNA precursors that are poorly converted to their mature forms may be polyadenylated and subsequently degraded. Furthermore other papers show that poly A can be important for RNase R degradation. Can you please further comment RNase R and polyadenylation?

We thank the reviewer for bringing our attention to this work. Indeed, we could envisage that addition of polyadenylation of stable structured RNAs could act as quality control pathway to make them substrates for RNase R. We now mention this in the discussion on page 13.

Referee #2 (Remarks to the Author):

Protein synthesis is an essential process in the cell, and requires delicate regulation for ribosome generation, maturation and degradation. Many structural studies have been focused in this field. However, structural insight into the degradation of ribosomes has been lacking. In this paper, the authors report cryo-EM structures of two distinct small ribosomal 30S subunit associated with the 3' to 5' exonuclease-RNase R in two distinct degradation intermediate states. One structure represents the initial binding of RNase R to 30S to facilitate degradation of the anti-Shine-Dalgarno sequence, and the other reveals a large conformational change both in 30S and RNase R from the initial position. These structures provide a mechanistic basis for RNase R-mediated 30S degradation and suggest that RNase R targets orphaned 30S using a dynamic anchored binding site switching mechanism. Overall, the topic is important and the structural data are solid with good quality. My major concern is the biochemical data and functional analyses, they do not match the quality of the structural data and do not provide enough supports as they should for the conclusions from the structures.

Comments:

1. In the paper, the authors mentioned the secondary structures of 30S many times, such as h40, h44 and h45. Given the broad range of the audience, there should at least be a schematic picture of the 30S RNA with all the major secondary structures labeled, so that readers could easily understand what are the authors' points in the figures.

As requested, we have added a schematic picture of the 16S secondary structure as panel f in Figure 2, highlighting the main rRNA helices that are relevant for this study.

Fig. 2 e and f. The authors employed in vitro degradation assay and biochemical mapping to find out the truncation site in the first degradation intermediate state. Since this is an in vitro purified system, can the authors isolate the products and use sequencing method to figure out the exact sequence of 30S in this state?

We have now performed high-throughput sequencing of rRNA samples both from the in vitro assay and from native pullouts, which is included in the manuscript as the new Expanded Data Figure 2 with associated text on page 8. The mapping of 3' ends clearly shows a main truncation at residue C1391 of the 16S rRNA. Both the in vitro assay and native pullouts are enriched in this site. Furthermore, we observe additional truncations of lower abundance in the native pullouts that are covering all of h28 (residues 1388 - 1412) (ED Figure 2). These results are in perfect agreement with our Northern Blot analysis shown in Figure 2e, as probe c (targeted at nucleotides 1372-1391) still produces a full signal, whereas probe d (targeted at nucleotides 1392-1409) already shows a significant reduction in the signal. Finally, probe e (targeted at nucleotides 1412-1432) loses the signal which coincides with the 3' boundary of our sequencing results. In addition, we believe that it is one of the longer truncations at h28 that is present in our cryo-EM reconstruction of State I, as we still observe density up to residue ~1397 at the head region of the 30S subunit (Fig. 2g). We note that during cryo-EM sorting, we excluded >1mio particles because they showed highly fragmented density for RNase R and the 30S head, whereas our State I (with three sub-states) amounted only to 89k particles. We therefore believe that the large portion of excluded particles would contain the shorter form of 16S rRNA ending around C1391 which might be more flexible in the cryo-EM analysis.

Figure 3. The authors used both in vitro degradation assay (a-e) and in vivo composition analysis to study the importance of RNase R. Given that the authors now have all the detailed structural information of the 30S-RNase R complex, they should design some key point mutations based on the structure to understand the functional important of the essential RNA-protein interfaces revealed by the structure. This would be much stronger than the current data which do not actually require the structural information. This is especially true for the case of Figure 3f, whose current form does not support the important value of the structure. And again, the authors could use sequencing method to find out the exact sequence of the products (Figure 3b 3d).

As requested, we have designed several RNase R mutants that target 16S rRNA interaction surfaces; we have created mutants in the HTH domain, the CSD2 domain and the DRP motif (RNB domain), as these contacted the 30S subunit in our structural model (Fig. 1). After native pullouts of these mutant proteins we observed changes in the RNA composition of associated 30S subunits (shown in a new panel in Fig. 3f) and described in the main text on page 9-10. The data shows that mutating both the HTH and RNB domain reduces the amount of truncated 16S rRNA observed. In addition, a double mutant of the RNB mutant with the CSD2 mutant mimics the single RNB mutant (Fig. 3f). We interpret this data as a result of weaker interaction of RNase R with 30S subunits, which could result in reduced binding strength at the beginning or during the initial turnover of h44+45. We agree that the analysis of subunit ratios during steady state growth did not fit well in this figure and have now presented the sucrose gradient analysis with new data in ED Figure 3 (as requested by reviewer 3).

Figure 1c. It is hard to identify the N and C termini. They have the same color.

We have now colored the domains of RNase R in Figure 1c according to the schematic in Figure 1b for clarity.

Figure 1d, CSD1 is not labeled.

We have added the requested label to Figure 1d.

Referee #3 (Remarks to the Author):

Dimitrova-Paternoga et al. describe among the first structural insights into ribosome subunit degradation, specifically *Bacillus subtilis* 30S rRNA interaction with RNase R. They perform cryogenic electron microscopy (cryo-EM) analysis of RNase R-30S complexes immunoprecipitated from *Bacillus subtilis* lysates via RNase R-FLAG overexpression, deriving two primary/global structural states by multiparticle sorting/refinement. These structures visualize direct RNase R interaction with truncated 3' 16S rRNA, in 30S states with unprecedented large-scale domain rearrangements, and lacking ribosomal proteins, both of which are distinct from previously solved mature 30S structures. They substantiate their structural data with biochemical analysis of in vitro mature 30S degradation by recombinant RNase R, and ex vivo 30S accumulation in an RNase R deletion (Δ rnR) strain. The authors ultimately conclude from these data that RNase R degrades mature 30S made vulnerable by the lack of complexation with mRNA and/or 50S subunits. The authors speculate in the Discussion that RNase R may degrade 30S subunits in conditions where its homeostasis is necessary, such as stationary growth, starvation, and/or quality control when the subunit is damaged or stalled during translation.

The Reviewer finds the study and its results of high interest, making a unique contribution to the ribosome field that has major gaps its understanding of ribosome degradation. The structural and biochemical data are solid, rationalizing a mechanism of 16S rRNA degradation by RNase R. The major issues to be addressed in revision principally concern the authors' interpretation of the data, with regard to the 30S states targeted by RNase R in vivo. In particular, it will be important for the authors to disentangle two possible interpretations of their data: the biological function of RNase R is to degrade mature 30S, and/or RNase R degrades immature 30S assembly precursors. The following are detailed comments geared towards clarifying these two possibilities, thus linking the excellent structural data to a biologically solid interpretation.

1.0. (major) The authors' interpretation of the structural data is that RNase R degrades "orphaned" (nice term) "mature" 30S, as summarized in the Discussion (lines 314-316; Figure 5). RNase R-30S structures are reported to be absent for ribosomal proteins bS21 (lines 135-136), uS2, and uS3 (lines 287-289). The absence of these ribosomal proteins is interpreted to be a consequence of RNase R-mediated displacement, or their dissociation is a prerequisite for RNase R binding (lines 136-138). However, complicating this interpretation is that precisely bS21, uS2, and uS3 are absent in the latest-stage 30S assembly intermediates (eg. assembly map from Chen & Williamson, JMB 2013, <http://dx.doi.org/10.1016/j.jmb.2012.11.040>, Figure 4a). This is substantiated by emerging work from the Davis and Ortega labs, which includes structural analysis of 30S assembly in the absence of KsgA, a biogenesis methyltransferase (<https://doi.org/10.1101/2022.07.13.499473>). It's notable in this second paper that bS21 and uS2 appear to be particularly affected in $\Delta ksgA$ cells, and that the map density for the 30S head is highly fragmented. The Reviewer appreciates that the biochemical data with purified mature 30S supports the idea that reconstitution with RNase R can degrade orphaned mature 30S in vitro – these data are solid. Nonetheless, the above issues raised in the overexpression-based ex vivo-derived structures is whether RNase R does degrade orphaned "mature" 30S in vivo. This issue is detailed further in several related Reviewer comments, as follows.

Our results suggest that any 30S particle with an accessible 3' end could be a substrate for RNase R, therefore, we also see no reason why very late pre-30S particles should not be subject to RNase R degradation. We thank the reviewer for raising this point and now mention this in the discussion on page 12. However, we would like to point out that while we think that late biogenesis intermediates could be substrates for RNase R, we have no evidence to suggest that State II is a late biogenesis intermediate with a pre-rotated head that was directly recognized and bound by RNase R. In the recent structures of late biogenesis intermediates where the head is disordered (e.g. Sun et al NSMB 2023), there is no evidence that this flexibility derives from the dramatic movements observed in State II of the RNase R-30S complex determined here. Rather, in these late biogenesis structures, it appears that helix 28 is intact and that the flexibility simply derives from a more limited head rotation. This contrasts with State II where we do not observe density for any part of h28, explaining why such a dramatic movement of the head is possible. Moreover, as the reviewer acknowledged, we observe the same RNA truncation in our native pullouts as seen in the in vitro assays using mature 30S subunits. From this, we conclude that RNase R encounters the same barrier for turnover of the bulk of 16S rRNAs in each sample, independent of the occupancy of late-binding ribosomal proteins and possible biogenesis factors.

Nevertheless, to investigate a potential role of RNase R in biogenesis, we have performed the requested analysis of the RNA from the 30S subunit peak in the RNase R deletion strain. This analysis revealed that the particles that accumulate in the absence of RNase R do not appear to contain pre-16S, but rather mature 16S rRNA, which leads us to favour a model where RNase R operates predominantly on mature 30S as evidenced from our *in vitro* data (and the presence of uS2 and uS3 in State I), but we leave the possibility open that mature as well as very late intermediates could be substrates *in vivo*. This is now mentioned in the discussion on page 12.

1.1. (major) Mass spectrometry was used to confirm co-immunoprecipitation of ribosomal particles with RNase R-FLAG (line 86), with the authors noting the enrichment of ribosomal proteins. Notably, in the related Supplementary Data table, many subunit biogenesis factors appear to co-purify with RNase R-FLAG at levels comparable to ribosomal proteins themselves (eg. CshA, YqeI, RnjB, RsfS, RnjA, RimM, RlmCD, TrmB, InfC, EngB, RsmB, DbpA, Obg). Complicating the picture is that many of the ribosomal proteins are 50S (rather than 30S), and some of these biogenesis factors are 50S-associated (eg. RsfS, InfC, DbpA, Obg). These observations are apparent when rank-sorting the table by the average iBAQ of RNase R-FLAG IP replicates, and also when rank-sorting the RNase R-FLAG average normalized to the iBAQ control. As the authors raise the “question as to how mature 30S subunits protect themselves from RNase R action in the actively growing cell” (lines 221-222), it is thus worth considering the action of biogenesis factors in protecting assembly intermediates, in addition to mRNA and the 50S.

We understand the reviewer’s point relating to the mass spectrometry data, however, we do not believe that the mass spectrometry results provide an accurate representation of RNase R-30S complexes being analyzed. This is mainly because we employ a rapid one-step purification to preserve as many novel ligand-ribosome interactions as possible using a clean, but not excessive, washing strategy during the pullouts which produces reasonably pure complexes for cryo-EM analysis (see Fig. S1a). Therefore, we do expect some level of background from our pullouts, which is evident from the detection of many 50S ribosomal proteins, including factors that interact with the 50S (such as RsfS, DbpA and Obg), despite the 50S/70S particles representing only a small (2%) proportion of the cryo-EM dataset. Similarly, the mass spectrometry data indicate the presence of many proteins that interact with the 30S subunit (ranging from initiation factor IF3 (InfC) to 30S biogenesis factor RimM), however, no additional density is observed for any of these factors in any of the 30S subpopulations within the cryo-EM dataset. We also note that in previous mass spectrometry analysis of native RqcH-50S complexes isolated in the same way as in this study, amongst several 30S ribosomal proteins, many additional protein factors such as EngB, RbgA, RluB, elongation factor EF-G and initiation factor IF2 were detected that were also not visualized in any subpopulations (Lytvynenko et al. Cell. 2019).

Thus, collectively, we are reluctant to draw conclusions from the mass spectrometry data, but at the same time we do not want to exclude the possibility that RNase R could operate on very late pre-30S particles, as mentioned in the discussion on page 12. In addition, we discuss the possible role of biogenesis factors in protecting the 16S rRNA 3’ end on page 13 and in Fig. S7d-f.

1.2. (major) Sucrose density gradients analyzed subunits in WT vs. Δ nrn cells (Fig. S7), with the interpretation that loss of RNase R leads to the accumulation of 30S relative to the 70S (Fig. 3f). The authors are encouraged to overlay the gradient profiles in Fig. S7 to facilitate direct comparison, since the y-axis limits are not consistent. Regardless, it would appear that while there is a legitimate accumulation of 30S (whether compared against 70S or 50S), the 70S itself appears decreased in the Δ nrn strain compared to WT. Given the questions regarding 30S assembly intermediates raised above, it may be that the accumulation of 30S represents 70S-incompetent precursors that would normally be degraded by RNase R. The consistent appearance of a peak with intermediate sedimentation between the 30S and 50S in both WT and Δ nrn begs the question of what these peaks actually represent. The authors should run an RNA gel derived from serially collected gradient fractions to evaluate the identity of the peaks. The authors should describe in the Methods at what growth stage the cells were collected before lysis in this experiment, as this additional peak may be subunit degradation during the transition from logarithmic to stationary phase (mediated by many RNAses), as described by the Murray Deutscher lab, for example. Finally, statistical significance testing of mean differences between quantified peaks must be presented in Fig. 3f (and legend), with individual replicate values and dispersion represented in the graph.

*We have repeated the sucrose gradient experiment and collected fractions from major peaks, as requested. The cells were grown, as before, to the transition from logarithmic to stationary phase ($OD_{600} = 1.4$), the same condition that was used for the pullouts that were analyzed structurally. The results including an overlay of gradient profiles are shown in the new Expanded Data Fig. 3 with associated text on page 10. As before, we see a slight but reproducible increase in the amount of 30S subunit relative to 50S subunits, as well as a slight decrease in 70S ribosomes in the RNase R deletion strain relatively to the wildtype strain. In addition, we again observe the peak of intermediate sedimentation between 30S and 50S subunits. As requested, we have now analyzed the fractions from all peaks on RNA gels and performed Northern blotting to probe for the presence of pre-16S species (ED Fig. 3c). As a control for pre-16S accumulation we have also extracted RNA from the *yqeH* deletion strain, which has been previously reported to accumulate pre-16S species with immature 3' ends in *B. subtilis* (Loh et al. *Genes & Genetic Systems*, 2007). As can be seen in the ED Fig 3c, there is no evidence for pre-16S in any of the peaks from the wildtype or RNase R deletion strain, whereas pre-16S species are seen to accumulate in the *yqeH* deletion strain. Although this would suggest the increase results mainly from mature 30S subunits, we cannot exclude that very late biogenesis intermediates with mature 16S rRNA also contribute – this is now mentioned in the text on page 10. We also report now in the methods that the cells were collected at late log phase ($OD_{600} = 1.4$), the same conditions used for cryo-EM.*

1.3. (suggestions) The authors are encouraged to address Comments 1.0-1.2 as they see fit with biochemical/phenotyping assays - the Reviewer does not expect further labor-intensive structural experiments. Some combination of the following experiments is suggested to disentangle the orphaned “mature” vs. “assembly precursor” 30S targets of RNase R. (1) Knock-in RNase R-FLAG, followed by IP-mass spectrometry to assess the binding partners in more physiologic conditions.

(2) Generate orphaned mature 30S by overexpression of a 70S splitting factor or anti-association factor (eg. ObgE, RsfS), which should accumulate free mature 30S, and see if RNase R is enriched or selectively associates with these abnormally accumulating 30S (eg. by sucrose gradient with RNA gel of the fractions). (3) Antisense-knockdown or knockout of biogenesis factors (eg. KsgA, RbfA) to generate precursor 30S, and likewise analyze if RNase R enriches with these precursors. (4) Culture in stress conditions or induce stalling, similar to the speculated conditions of RNase R-mediated 30S homeostasis described in the Discussion, where RNase R should be recruited to specifically mature vs. precursor 30S. Performing RNase R-IP at earlier growth time points, such as early log phase (OD₆₀₀= 0.3), or mid log phase (OD₆₀₀= 0.5), may also be informative. Of course, it may be that ultimately RNase R targets both mature and precursor 30S, but these data would critically facilitate interpretation of the underlying 30S structures and physiologic role of RNase R.

As mentioned in the following point (2.0), the yields from the pull-downs using a strain with a knock-in of RNase R-FLAG in an endogenous locus are very low, therefore, this approach could not be used for preparing samples for structure determination. Nevertheless, we agree with the reviewer that it would be important to analyze the pulldown from the integrated RNase R-FLAG and compare it with the sample from the plasmid expressed RNase R-FLAG. Given the background that is observed in the pulldowns using mass spectrometry (see point 1.1 above), we have instead focused on analyzing the sample using RNA gels and Northern blotting. In the new Supplementary Fig. 1c,d we demonstrate that the RNA pattern for the sample from the integrated RNase R-FLAG appears to be very similar, if not identical, to that observed for the sample from the plasmid expressed RNase R-FLAG. In particular, we observe the truncated 16S product to be present in high amounts in both samples, as seen in Supplementary Fig. 1d. As mentioned above (point 1.2), we have also used Northern blotting to look for accumulation of pre-16S in the wildtype versus RNase R deletion strain, using the yqeH deletion strain as a positive control. As seen in the new Expanded Data Fig. 3c, we observe no accumulation of pre-16S in the wildtype or RNase R deletion strain, whereas clear pre-16S is observed in the positive control using the yqeH deletion strain. As mentioned above, we nevertheless think it is possible that very late pre-30S particles could be a substrate for RNase R and mention this in the discussion on page 12.

2.0. (minor) The authors' approach to purify RNase R-ribosome complexes for cryo-EM appears to involve RNase R-FLAG overexpression in *Bacillus subtilis* with an IPTG-inducible ectopic expression vector (pHT01), rather than an alternative approach such as knock-in to the native locus. This might not be immediately clear to a non-specialist reader in the Results or Methods, and should be explicitly described with more detail, warranted in the Methods especially, such as: What is IPTG protocol for expression of pHT01? At what temperature and shaking are the cells grown?

The reviewer is correct that RNase R expression from the endogenous locus is very low, hampering cryo-EM analysis. We now provide a comparison of knock-in expression to our plasmid-based expression (Figure S1c). We would like to highlight that no IPTG-inducible expression was used in our experiments. We based our constructs on a pHT01-derivative in which the inducible pGrac promoter had been replaced with a constitutive p43 promoter and a weak ribosome-binding site. Additionally,

we have performed a large-scale purification of the knock-in strain which yielded enough RNA for a comparison gel (Figure S1d); This analysis shows that plasmid-based purification is indistinguishable from the knock-in and therefore suitable for cryo-EM analysis (Figure S1d). Furthermore, we added the request information (Growth at 37 °C, shaking at 145 rpm) to the Methods section.

2.1. (minor) The purification approach (Comment 2.0) for structural analysis impacts the interpretation of the structural data. The Reviewer applauds the ex vivo-derived approach, and appreciates that overexpression of RNase R is likely to be helpful, maybe even critical, to enrich sufficient material for cryo-EM (rather than physiologic expression from the native locus). However, the potential complication is that super-stoichiometric RNase R may accumulate in a non-physiologic manner on complexes normally protected robustly by biogenesis factors. Indeed, then the physiologic role of RNase R may be to act on orphaned mature 30S, as the authors expect from their biochemistry, but the specific approach could derail this interpretation with structures potentially representative of late-stage assembly precursors. Thus, addressing this limitation in the Discussion, in addition to addressing the major points above, would clarify the extent to which these data reflect RNase R's physiologic role.

The reviewer is quite correct that it was necessary to overexpress RNase R in the cell to obtain sufficient amounts for structural analysis. We now include data in Figure S1 illustrating the low level of endogenous RNase R expression. We also performed pulldowns using FLAG-tagged RNase R from the endogenous locus. Although the yield was low, it was sufficient to analyze the immunoprecipitated RNA using acrylamide gels. We now include this analysis in Figure S1 panel d to illustrate that although the yields are lower, the banding pattern and importantly the presence of the truncated 16S rRNA appear to be identical. Nevertheless, we agree that the overexpression could lead to non-specific interactions and now mention the experimental conditions in the text on page 4.

2.2. (minor) The Methods description of the mass spectrometry is vague as to what samples were analyzed. It would appear from the table that purified 30S and 70S (presumably purified by sucrose gradients?) were shot for comparison. Clarification in the Methods text is required.

We apologize for the confusion. The 30S and 70S subunits were analyzed for a different experiment and the data was left in the table by mistake. For this study we only analyzed the RNase R pullouts and the corresponding vector control.

2.3. (minor) The authors are encouraged to cite Failmezger et al. 2016 (<https://doi.org/10.1371/journal.pone.0168764>), which supports their interpretation regarding helix 44 and targeted 30S degradation.

We thank the reviewer for this suggestion. Indeed, the findings of Failmezger et al that the 30S was more susceptible to degradation than the 50S and that active translation seemed to prevent ribosome degradation to some extent are very relevant and we now mention this on page 13.

Reviewer Reports on the First Revision:

Referees' comments:

Referee #1 (Remarks to the Author):

Structural insights into the degradation of ribosomes have been lacking, and in this manuscript the authors report the native structures of two distinct small ribosomal 30S subunit degradation intermediates from *B. subtilis* associated with ribonuclease R. The mechanism proposed indicates that RNase R targets orphaned 30S subunits using a dynamic anchored binding site switching mechanism.

The manuscript is substantially improved in its revised version and I believe that the work can now be accepted for publication.

Referee #2 (Remarks to the Author):

The authors have successfully addressed all the points raised by the reviewers and the revised manuscript is suitable for publication.

Referee #3 (Remarks to the Author):

The authors provide an exemplary revision, sufficiently addressing my comments. I support the publication of this manuscript in its current form.